# Long-term monitoring of SARS-CoV-2 seroprevalence and variants in Ethiopia provides prediction for immunity and cross-immunity

Simon Merkt [1,15], Solomon Ali [2,15], Esayas Kebede Gudina [3,15], Wondimagegn Adissu [3], Addisu Gize[2,4], Maximilian Muenchhoff [5,6], Alexander Graf [7], Stefan Krebs [7], Kira Elsbernd [8,9], Rebecca Kisch [8], Sisay Sirgu Betizazu [2], Bereket Fantahun[2], Delayehu Bekele [2], Raquel Rubio-Acero[8], Mulatu Gashaw [3], Eyob Girma [3], Daniel Yilma [3], Ahmed Zeynudin[3], Ivana Paunovic [8,10], Michael Hoelscher [6,8,10,11], Helmut Blum [7], Jan Hasenauer [1,12,13,16] ✉, Arne Kroidl [6,8,16] ✉ & Andreas Wieser[6,8,10,14,16] ✉

Under-reporting of COVID-19 and the limited information about circulating SARS-CoV-2 variants remain major challenges for many African countries. We analyzed SARS-CoV-2 infection dynamics in Addis Ababa and Jimma, Ethiopia, focusing on reinfection, immunity, and vaccination effects. We conducted an antibody serology study spanning August 2020 to July 2022 with five rounds of data collection across a population of 4723, sequenced PCR-test positive samples, used available test positivity rates, and constructed two mathematical models integrating this data. A multivariant model explores variant dynamics identifying wildtype, alpha, delta, and omicron BA.4/5 as key variants in the study population, and cross-immunity between variants, revealing risk reductions between 24% and 69%. An antibody-level model predicts slow decay leading to sustained high antibody levels. Retrospectively, increased early vaccination might have substantially reduced infections during the delta and omicron waves in the considered group of individuals, though further vaccination now seems less impactful.

The COVID-19 pandemic continues to have a significant global impact, with a substantial number of deaths continually being recorded worldwide (covid19.who.int). However, observations indicate a shift from the initial phase of the pandemic to an endemic stage, with reduced confirmed case numbers as well as deaths. Despite this, the emergence and evolution of more transmissible variants still pose a threat globally, necessitating ongoing monitoring by organizations such as the World Health Organization (WHO). In order to better prepare for future Sars-CoV-2 waves and potential pandemics, understanding the dynamics of the disease and the immune response protecting against infection as well as severe disease courses is crucial.

Policymakers rely on accurate data to inform vaccination strategies and intervention measures. However, these strategies may differ greatly depending on circumstances like information about the actual virus spread and public acceptance of policies. Especially within the African continent, comprehensive data is scarce. Even in July 2023, the

WHO still only lists 9.5 million confirmed cases of SARS-CoV-2 in the whole of Africa. Given our data, serological evidence of past infection in Ethiopia alone suggests that by autumn 2022, there were ten times as many infections in Ethiopia as officially reported[1].

Besides the scarcity of data, African countries, including Ethiopia, face unique challenges in dealing with the pandemic, such as limited testing infrastructure[2], insufficient vaccine supplies[3], low vaccine acceptance[4], and being overlooked in global research efforts[5]. For Ethiopia in particular, research shows that though adequate pandemic prevention strategies have been enacted over time, shortages of medical supplies and equipment is an ongoing struggle[6].

In 2021, we demonstrated a severe under-reporting of COVID-19 cases in Ethiopia through an antibody prevalence study[1]. By employing epidemiological modeling, we predicted prevalence levels above 50% for the population. While this earlier phase of the pandemic has received some research attention, later phases of the SARS-CoV-2 pandemic, including the Delta and Omicron waves, remain inadequately investigated in Ethiopia[7,8]. Additionally, due to very limited access to sequencing facilities, the knowledge about circulating variants has been scarce. Previous publications touch upon this topic hypothetically, e.g. Gudina et al. by simulating a scenario with two variants[1], but longitudinal data on variant distribution has only recently become available for Ethiopia[9]. We have simultaneously acquired broad data to address the gaps for modeling and prediction of the epidemic in Ethiopia.

In this study, we obtained sequencing results for SARS-CoV-2 samples collected at various time points between October 2020 and July 2022 at two different sites in Ethiopia. This dataset enabled us to investigate the composition of variants of concerns (VOCs) between the initial appearance of COVID-19 in Ethiopia in March 2020 to the spread of Omicron variant BA.4/5 as the dominant genotype in fall 2022. Additionally, we extended our serology-based antibody survey by conducting two further sampling rounds to cover the time span between late fall 2020 to April 2022 in a total of five sampling rounds. In addition to the serological testing against Anti-nucleocapsid antibodies (Anti-N), all samples were re-tested against anti-spike antibodies (Anti-S), and questionnaires were used to explore vaccination- and potential infection status for all participants. Using this large and multidimensional dataset for analysis, we developed a large-scale multivariant model to characterize the infection pathways and to explore the cross-immunity properties among different variants circulating in Ethiopia. This analysis allowed us to gain insights into the interplay between the variants and their impact on the overall population's immune response.

Furthermore, we leveraged the information from multiple rounds of sampling, which provided Anti-N and Anti-S antibody levels of individuals. The resulting dataset was used for a detailed temporal analysis, comparing the antibody levels observed during the initial three rounds with those from the subsequent two rounds. We utilized a second epidemiological model to predict future antibody dynamics, providing insights into the expected long-term immunity landscape in the Ethiopian population. This might provide decision makers with information which is helpful for the assessment of the situation and the choice of appropriate measures.

In summary, this study expands upon previous findings and presents novel insights into the antibody dynamics and concurrent variant prevalence in Ethiopia. By integrating modeling techniques and broad datasets, we aim to contribute to a deeper understanding of SARS-CoV-2 infections and the implications for public health interventions and vaccination strategies in Ethiopia, other resource-limited settings, and beyond.

## Results

### Antibody data reveals majority had multiple infections

In our previous study, we assessed the dynamics of COVID-19 infection between August 2020 and April 2021 in Addis Ababa and Jimma, Ethiopia[1]. To understand how the COVID-19 pandemic evolved afterwards, we conducted two additional rounds of sampling. As our previous study predicted a complete transmission within the population for SARS-CoV-2 in Ethiopia by late 2021, we complemented the previous semi-quantitative analysis of Anti-N antibody levels by a quantitative analysis of the Anti-S antibody levels in the newly collected and historic samples to gain more detailed insight into possible reinfection occurrences. An overview of the demographics of the participants of the original three rounds and the two follow up rounds is shown in Table 1 (for healthcare workers Supplementary Table 1). Study flows are depicted in Supplementary Fig. 1.

Our SARS-CoV-2 specific antibody tests revealed that in April 2022, the majority of individuals (in Round 5: 95.9% of the healthcare workers and 94.8% of the community members), reacted positive for both Anti-S and Anti-N antibodies (Fig. 1a–e), suggesting an infection event. Based on a previous study, this result is unlikely to be explained by cross-reactivity[10]. In Round 3 (April 2021, Fig. 1c) and four (August 2021, Fig. 1d), significant numbers of samples were observed which showed isolated positivities for Anti-N or Anti-S. This can be explained by a delayed onset of either Anti-N or Anti-S response shortly after or during infection or, for Anti-S positivity, by vaccination. As large-scale vaccination campaigns started in Ethiopia rather late in November 2021, the data suggests that sampling in Round 3 coincided with waves of SARS-CoV-2 infections. First confirmed vaccinated individuals show up only in rounds four (August 2021, Fig. 1d) and five (April 2022, Fig. 1e). Interestingly, although the vaccines used in Ethiopia only induce Anti-S, most individuals vaccinated also showed reactivity for Anti-N (in Round 5: 94.8% of the healthcare workers and 96.4% of the community members), suggesting they had been exposed to the infection prior to or shortly after vaccination. By Round 4 all vaccines

**Table 1 | Demographic characteristics of community members participating in study**

| | Jimma | | | | | Addis Ababa | | | | |
|---|---|---|---|---|---|---|---|---|---|---|
| | R1 (Dec 20) | R2 (Jan 21) | R3 (Feb 21) | R4 (Aug 21) | R5 (Apr 22) | R1 (Jan 21) | R2 (Feb 21) | R3 (Apr 21) | R4 (Sep 21) | R5 (Mar 22) |
| Participants | 536 | 325 | 267 | 539 | 575 | 361 | 314 | 721 | 424 | 461 |
| Age | 30 (19, 63) | 30 (19, 62) | 32 (19, 63) | 33 (20, 65) | 32 (19, 63) | 36 (21, 68) | 36 (22, 67) | 35 (21, 67) | 33 (19, 65) | 38 (20, 68) |
| Sex | | | | | | | | | | |
| Female | 260 (48.5%) | 166 (51.1%) | 136 (50.9%) | 331 (61.4%) | 317 (55.1%) | 279 (77.3%) | 236 (75.2%) | 360 (49.9%) | 209 (49.3%) | 162 (35.1%) |
| Male | 276 (51.5%) | 159 (48.9%) | 131 (49.1%) | 207 (38.4%) | 258 (44.9%) | 79 (21.9%) | 70 (22.3%) | 109 (15.1%) | 71 (16.7%) | 299 (64.9%) |
| Missing | 0 (0.0%) | 0 (0.0%) | 0 (0.0%) | 1 (0.2%) | 0 (0.0%) | 3 (0.8%) | 8 (2.5%) | 252 (35.0%) | 144 (34.0%) | 0 (0.0%) |
| Anti-N positive | 139 (25.9%) | 114 (35.1%) | 107 (40.1%) | 313 (58.1%) | 543 (94.4%) | 165 (45.7%) | 150 (47.8%) | 234 (32.5%) | 286 (67.5%) | 458 (99.3%) |
| Vaccinated | 0 (0.0%) | 0 (0.0%) | 1 (0.4%) | 47 (8.7%) | 195 (33.9%) | 0 (0.0%) | 0 (0.0%) | 0 (0.0%) | 28 (6.6%) | 167 (36.2%) |

Age denoted as median and 90% quantiles, and sex in absolute and relative numbers. Round 1-3 (R1-R3) are the previous study[1].

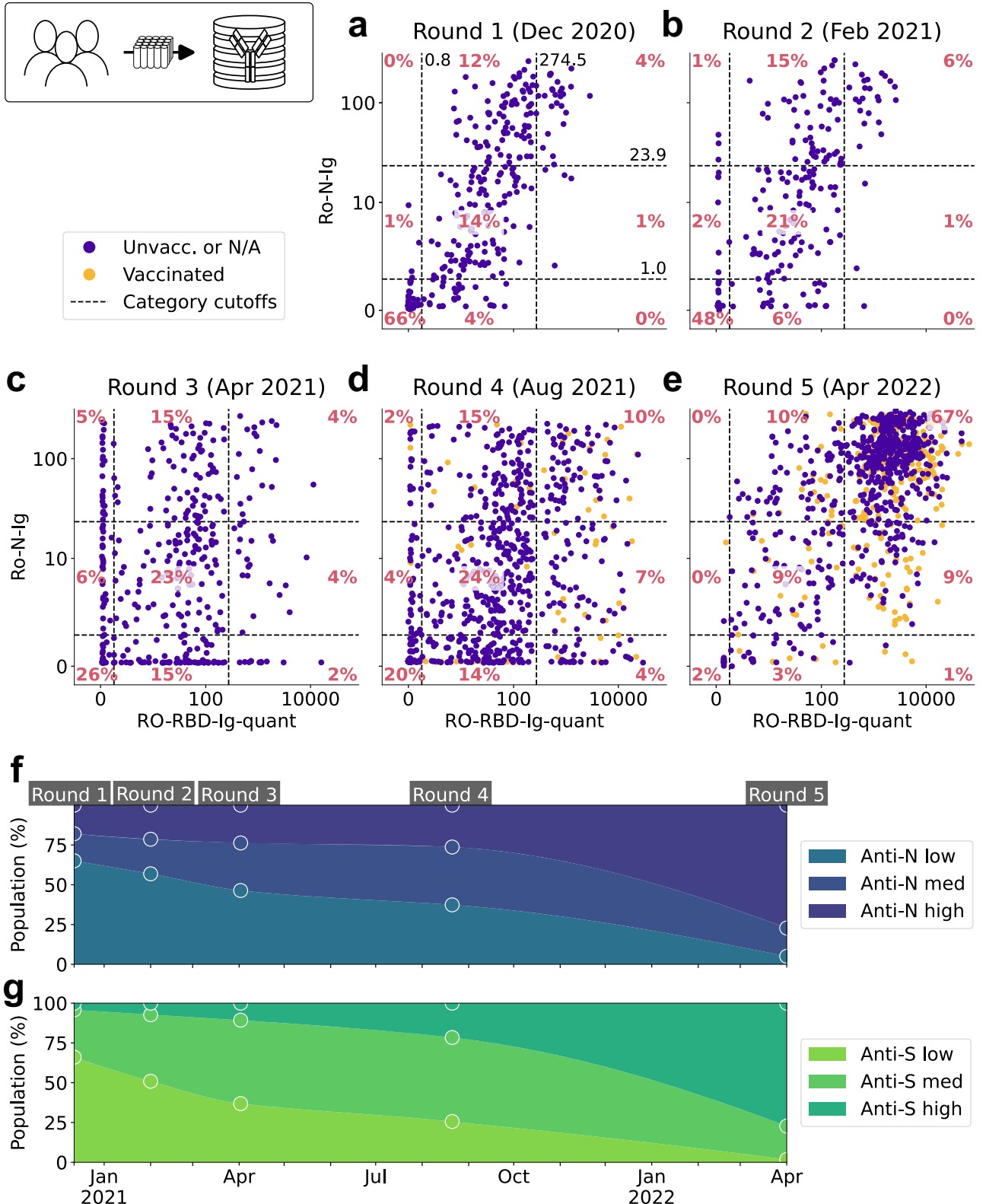

**Fig. 1 | Ro-N-Ig and Ro-RBD-Ig-quant measurements of five rounds of convenience sampled community members. a**–**e** Scatterplots displaying the relationship between levels of N- and S-specific antibodies across five rounds of measurement. Known vaccination status of each participant indicated by colors, cutoff levels indicated by dashed lines and percentages of people per category annotated in red. **f**–**g** Antibody levels over time between end of 2020 and April 2022. The observations are indicated by circles and the trend is indicated via smoothing splines constructed on the basis of these data. Source data are provided as a Source Data file.

in Ethiopia were Covishield (AstraZeneca type vaccine manufactured by Serum Institute of India) and by Round 5 Johnson & Johnson has become another major type of the vaccine. Although few doses of Sinavac/Sinopharm, Sputnik-V, Moderna, and Pfizer-BioNTech were reported to be donated to the country, they were very little and hence negligible. Therefore we can safely disregard the influence of mRNA vaccines in our study.

Analyzing the magnitude of the Anti-S responses considered positive (above the test threshold of 0.8), we observed two populations, separating positive samples into those with higher and lower levels (Fig. 1c–e, Supplementary Fig. 3c–e, Fig. SN1). Comparing the data in this study and with experience gathered in our population-based studies in Munich, Germany[11], it can be appreciated that one exposure to SARS-CoV-2 with a natural infection generally induces Anti-S values below a cutoff value centered in the middle of the antibody level range (shifted log-scale) as indicated with the vertical dashed line in Fig. 1a and b. Higher Anti-S levels are only reached after multiple exposures leading to a boosting effect. Employing 1-dimensional k-means clustering with two means on the S-positive samples from all five rounds, we determined the cutoff value for the groups with one or multiple exposures to be 274.5 (for more details see Supplementary Information's Supplementary Note 1 and Fig. SN2). Anti-S results are diluted and measured within the linear range to provide quantitative results for all samples as described in more detail in the Methods section.

For Anti-N values, a clear division into two populations is not as evident as for Anti-S, likely due to the semi-quantitative nature of the Anti-N measurements. A noticeable shift towards higher Anti-N values is observed between Round 4 (Fig. 1d) and Round 5 (Fig. 1e). However, we also performed k-means clustering on Anti-N values to determine distinct categories, similar to the process carried out for the Anti-S signals. Using the calculated cutoffs and positivity thresholds, we assigned the individual patients for each round into the categories *low* (negative, i.e. below threshold), *medium* (positive, i.e. above positivity threshold but below calculated category cutoff), and *high* (above category cutoff) for both Anti-N and Anti-S, respectively.

Moreover, we summarized the progression of Anti-N and Anti-S level categories separately over time (Fig. 1f–g). Remarkably, in the latest round of sample collection in April 2022, a substantial proportion (75-80%) of the sampled individuals exhibited *high* antibody levels for Anti-N as well as Anti-S. Since Anti-N is only induced after an infection due to the spike-protein nature of the vaccines used in Ethiopia, this suggests that a significant fraction of the population had already experienced at least two exposures for each antigen by that time.

## Variant sequencing identifies all major substrains

The antibody data provide information about previous infections, but not about the SARS-CoV-2 variants which caused them. Moreover, up until very recently, there was no available data on virus variants in Ethiopia[9]. Hence, to better understand the pandemic, we sequenced a total of 1873 SARS-CoV-2 reverse transcription polymerase chain reaction (RT-PCR) positive swabs, collected in Jimma and Addis Ababa, between October 2020 and July 2022. Overall 574 sequences were of sufficient quality to allow full pangolin strain matching and were thus used for analysis.

The sequencing data revealed the presence of several variant strains, including wildtype (A and all without any "interesting" mutations, details below), wildtype* (B.1.480), alpha (B.1.1.7), beta (B.1.351), eta (B.1.525), delta (B.1.617.2 and AY.*), and the two omicron lineages BA.1 and BA.4/5 (Fig. 2a). At the beginning of the sampling period in autumn 2020, the wildtype strain was predominant (as expected) and accompanied by a notable presence of the wildtype* (B.1.480) strain. However, in late 2020 to January 2021, the alpha variant emerged and rapidly became the dominant strain, accounting for approximately

80% of the PCR-positive swabs by April 2021. During this time, the eta lineage also briefly appeared, which was previously reported as the predominant strain in Nigeria in early Spring 2021 (B.1.525 on cov-lineages.org). In Ethiopia, the eta lineage was unable to outcompete the alpha variant, and with the appearance of the delta variant in July 2021, both alpha and eta disappeared. In early 2022, the omicron BA.1 variant emerged and completely took over. Despite that we had only limited samples during the transition phase, it is evident that by June 2022, the BA.1 variant was subsequently substituted by omicron BA.4/5. The full and detailed results of the sequencing analysis can be found in the supplementary materials (Table SN1).

The mutational variety observed in our dataset is extensive, with mutations spanning from less than 10 to more than 90 mutations relative to the original wildtype variant that originated in Wuhan (Fig. 2b, c). As variations in the spike protein play a critical role for immune escape, we assessed this in more detail following the definition and mapping of outbreak.info's mutations of interest or concern (MOIC)[12,13]. For the observed strains, the presence and absence of MOICs are indicated in Fig. 2d. In previous studies[14–16], the overall number of mutations (Fig. 2b, c) was used as a measure for reinfection potential. Grouping our variants by MOIC allows us to maintain the statistical power of the lineage groups for subsequent analysis of potential cross-immunity while still retaining their relevant spike protein differences. The grid of distances of MOIC between observed lineage groups in Fig. 2e demonstrates that our dataset encompasses a range of distances up to 6, indicating diverse genetic distances between the variants. Moreover we see that our data set consists of variants which emerged earlier in other parts of the world, hence implies a continuous introduction of new variants to Ethiopia rather than a mutation of the wild-type inside of Ethiopia. We provide more information about these distances in the methods section of this paper.

## Multivariant model describes antibody prevalence and strains

The long-term antibody and variant data from Addis Ababa and Jimma provide valuable information about the course of the pandemic. Yet, the observations themselves did not allow for a direct assessment of infection or reinfection risk, or (cross-)immunity. Challenges are: (i) most study participants contributed to less than three of five rounds of antibody testing and (ii) the participant groups for antibody testing and swab collection were disjoint. Therefore, it is not possible to map the data types to each other and to analyze individual disease history. To achieve a good understanding of the COVID-19 dynamics and the interactions of different variants in Ethiopia, we instead employ epidemiological modeling of population averages.

We constructed a multivariant model to investigate the temporal evolution of the SARS-CoV-2 pandemic in Ethiopia. The model accounts for different sequences of infections and vaccination events (Fig. 3a). The sequence of infections and vaccinations - to which we refer in the following as *pathways* - is tracked to determine the immunity status of individuals. Each infection follows the SEIR schematic, with individuals transitioning from being susceptible to exposed, then infected, and finally recovered. Due to official vaccine availability in Ethiopia only after Round 3[17] in combination with our previous observation that vaccinated individuals are more likely to answer questions on the vaccination status on the questionnaire than unvaccinated individuals, we considered individuals without an answer ("N/A") as "unvaccinated" for modeling. The structure of the multivariant model is outlined in Fig. 3a using a small number of possible pathways. The model has a total of 364 possible pathways, and possesses 950 compartments and more than 950 transitions.

We allowed for immunity and cross-immunity conferred by previous infections and vaccination in the multivariant model. As the precise dependencies are not known, we assumed a variant-specific risk reduction for reinfections with previously encountered variants.

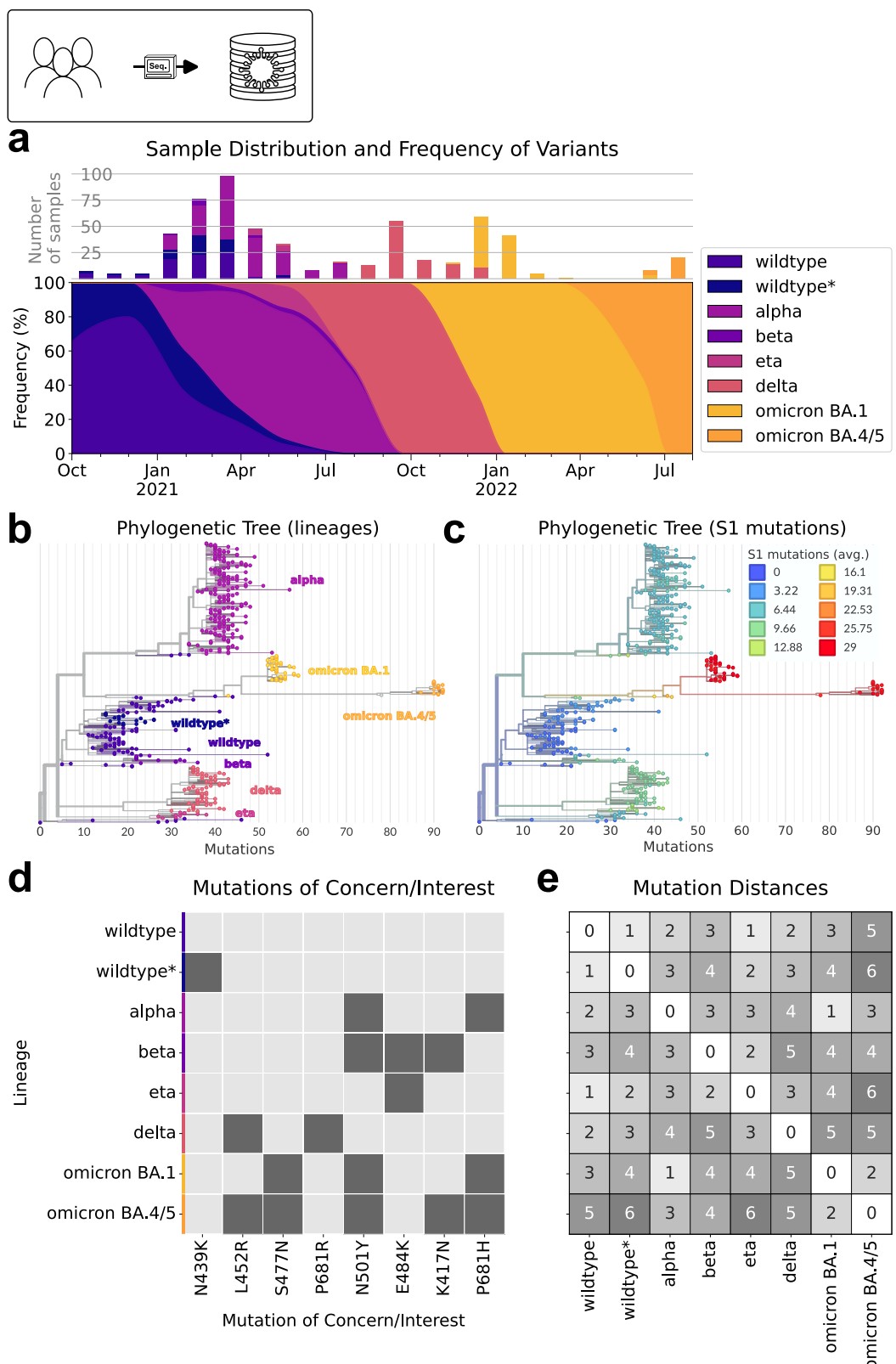

**Fig. 2 | Sequencing results of samples obtained between October 2020 and July 2022. a** Number of successfully sequenced samples, variant frequency and smoothed variant time-course. Variants are indicated using colors. **b,c** Phylogenetic tree of the sequenced samples, illustrating the relationships between variants and their sub-variants (full list of variants in Supplementary Information Table SN1). Distance between variants represented by overall difference in their mutations. Lineage groups **b** and number of mutations in the spike protein **c** highlighted by color. **d** Heatmap indicating which variants possess specific mutations of interest on their S1 protein. **e** Heatmap depicting MOIC mutation distances with respect to mutations of interest between different variants. Distance indicated by gray scale. Source data are provided as a Source Data file.

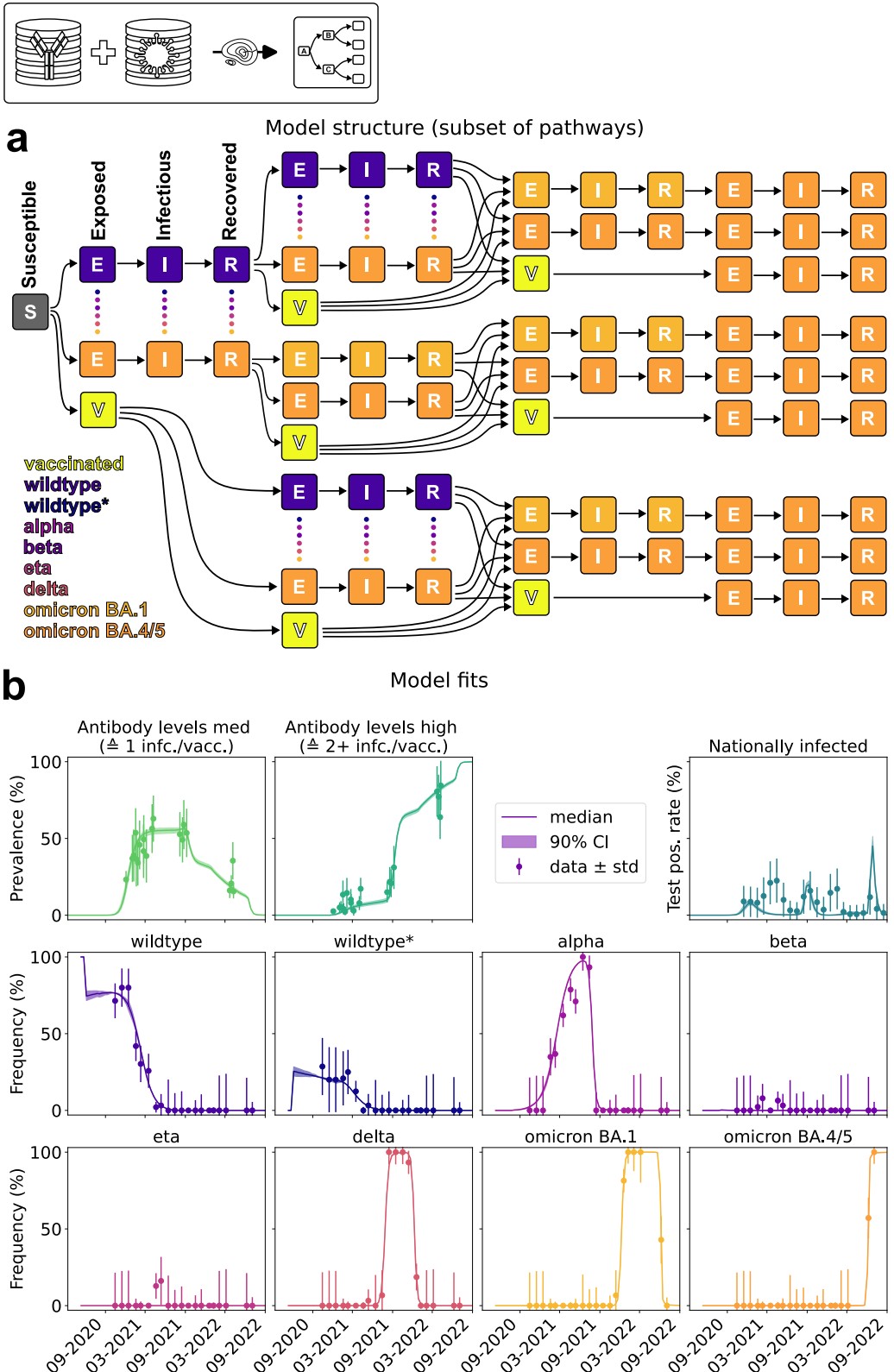

**Fig. 3 | Structure and fitting results of the multivariant model. a** Model structure depicting up to four consecutive infections/vaccinations. Potential infection pathways labeled by the stages S(usceptible), E(xposed), I(nfectious) and R(ecovered) and their respective variants highlighted by different colors. Only a small subset of the in total more than 350 possible paths is shown. **b** Model fitting results shown by progression of all observables against their respective (mean) measurements. Bayesian 90% credibility intervals for model simulation obtained by sampling included as well as the standard deviation of the measurements. Prediction simulations performed on $n = 6001$ parameter samples after burn-in from Markov chain Monte Carlo. Sample sizes of data points provided in Supplementary Note 2 (Table SN4). Source data are provided as a Source Data file.

For infection with a different variant, we assume that the infection risk depends on the difference of MOIC between the previously encountered variant and the variant to which individuals are exposed. In the case of multiple previous infections, the union of mutations from the previous variants is considered, and the distance to the new variant is calculated. This is based on the assumption that antibodies against regions with different MOIC can be developed. Vaccination is treated as a recovery from the wild-type infection. Exposure risk is also influenced by seasonality, which is incorporated using a 1-year-periodic factor. The unknown parameters of this seasonality factor and cross-immunity are estimated, along with the appearance times of the variants, incubation and recovery times, a basic exposure rate, and the exposure multipliers for the variants. A detailed mathematical description of the multivariant model and a complete list of its parameters is provided in Supplementary Information (Supplementary Note 2 and Table SN5).

To assess the evolution of the SARS-CoV-2 pandemic in Ethiopia, we parameterized the multivariant model using data on antibody levels, viral variant distribution, and national test positivity rate. The Anti-S antibody measurements were used to provide information on the fraction of individuals with a single infection or vaccination (medium level) and the fraction of individuals with at least two infections, vaccinations or a combination of both (high level). Since it is impossible to distinguish between vaccinations and infections from Anti-S levels we implemented observables corresponding to the medium and high levels without discriminating between vaccination or infection (c.f. Supplementary Note 2 for detailed equations). The viral variant data provided information on the relative levels of each of the eight variants, mapping the relative measurements to the percentage of individuals in an infectious state associated with each variant. The national PCR test positivity rate was used to determine the percentage of currently infected individuals, irrespective of the variant.

The parameterization of the model was performed using Markov chain Monte Carlo sampling. The sampling results revealed good agreement of the parameterized multivariant model with the observed data (Fig. 3b). The antibody levels and variant distributions (the primary focus of our investigation) are captured accurately. The national test positivity rate is described well up to two peaks (which might be caused by different regions in Ethiopia). In fact looking at the timing of the first peak, which is missed by our model, we see that our antibody data is already saturated and hence tells a different story than the nationally reported data. Most of the model parameters are well determined (Table SN5 in Supplementary Information) and in agreement with estimates provided in the literature. For a comprehensive description of estimation and uncertainty analysis results for specific parameters, as well as convergence information, we refer readers to the supplementary materials.

Overall, comparison of model simulation and data revealed that the proposed multivariant model provides a good description for the progression of the SARS-CoV-2 pandemic in Ethiopia. Furthermore, the assumed model for (cross-)immunity appears appropriate to accurately describe the data for Addis Ababa and Jimma.

### Reconstruction of infection history and cross-immunities

As the multivariant model provides an accurate description of the observed data, we used it to study the population-level infection history in Addis Ababa and Jimma. This infection history is encoded in the time-dependent state of the parameterized model, which is informed by our broad datasets.

The analysis of the model predicted that the most common pathway of infections and vaccinations was: 1st infection with wildtype, 2nd infection with delta, vaccination, and 3rd infection with omicron BA.4/5 (Fig. 4a, b). In particular wildtype*, alpha, beta, eta, and omicron BA.1 are not part of it, of which omicron BA.1 appears in the second most common pathway (delta, omicron BA.1, omicron BA.4/5) and alpha appears in the third most common pathway (alpha, delta, vaccination, omicron BA.4/5). The estimates indicate that a median of 12.7% with 90% credible interval (CI) of (10.9%,14.4%) of the inhabitants of Addis Ababa and Jimma followed this pathway. As suggested by the low percentage of individuals following the most common pathway, there has been a large degree of pathway variability. Indeed, the 10 most common pathways account for only 59.0% (42.8, 69.8) of the overall pathways (Fig. 4b). The high variability is caused by a large number of different combinations of virus variants. Overall wildtype, delta, and omicron BA.4/5 variants are the primary contributors to the infection progression (Fig. 4c). They are followed by wildtype*, alpha, and omicron BA.1, which also exhibit notable contributions. The model predicts a negligible impact of beta and eta variants, which is consistent with the data used to parameterize it.

The analysis of the time of infections (Fig. 4c) indicates three distinct waves, which coincide with reports for wildtype, delta and omicron BA.4/5. Notably, the emergence of the delta variant marks a shift where second infections start playing a significant role, which aligns with findings from other published studies[18]. Furthermore, with the introduction of the omicron variants, third infections become more prevalent, resulting in nearly the entire population experiencing at least two infections. Until September 2022, the occurrence of fourth infections appears to be minimal, likely due to the influence of vaccination and pre-existing immunity.

To assess the impact of cross-immunity on the pandemic, we assessed the corresponding model parameters used to describe it (Fig. 4d, e). The statistical inference suggests that the reinfection risk with the same variant - corresponding to a MOIC mutation distance of 0 - is reduced to 10.0% (5.1, 14.7) of the risk of an initial infection. In contrast, reinfection with different variants demonstrates a range of probabilities, ranging from 24.5% (21.3, 27.8) for a MOIC mutation distance of 1 (e.g., wildtype to wildtype*) to 68.6% (63.2.3, 72.4) for a distance of 6 (e.g., wildtype* to omicron BA.4/5). The 90% CIs for all variant-variant combinations are displayed in Supplementary Fig. 2.

Overall, the multivariant model provided insights in the infection history by linking datasets collected for different groups of individuals at different time points. Based on this, it sheds light on the differential susceptibility to reinfection based on the genetic distances between variants.

### Antibody-level model predicts high immunity and slow decline

The multivariant model enabled the assessment of the Anti-S antibody and variant data, yet, it is unable to fully exploit the comprehensive assessment of Anti-N and Anti-S antibody levels (Fig. 1a–e) available for a large fraction of our cohort. As this is necessary to assess waning immunity and the impact of vaccination rates, we decided to develop a tailored model for the analysis of these aspects.

We constructed an antibody-level model describing the dynamics of the Anti-N and Anti-S antibody levels. Following the analysis of the measurement data (Supplementary Information Fig. SN1), we implemented a discretization of both antibody levels in *low (negative)*, *medium* and *high*, which yielded a model with 9 state variables (Fig. 5a). Thresholds for these categories were inferred from the data (cf. antibody subsection of Results section and Supplementary Note 1 of Supplementary Information). Infections are assumed to result in increases of Anti-S and Anti-N antibody levels to the next higher category, while vaccinations are assumed to result only in an increase of Anti-S antibody levels to the next higher category. To account for the semi-quantitative nature of Anti-N measurements and the possibility of boosting Anti-N to *high* levels with a single infection, the model allows for a fraction of individuals in the Anti-N *low* category to directly transition to the *high* category. Antibody waning results in a shift to a lower category.

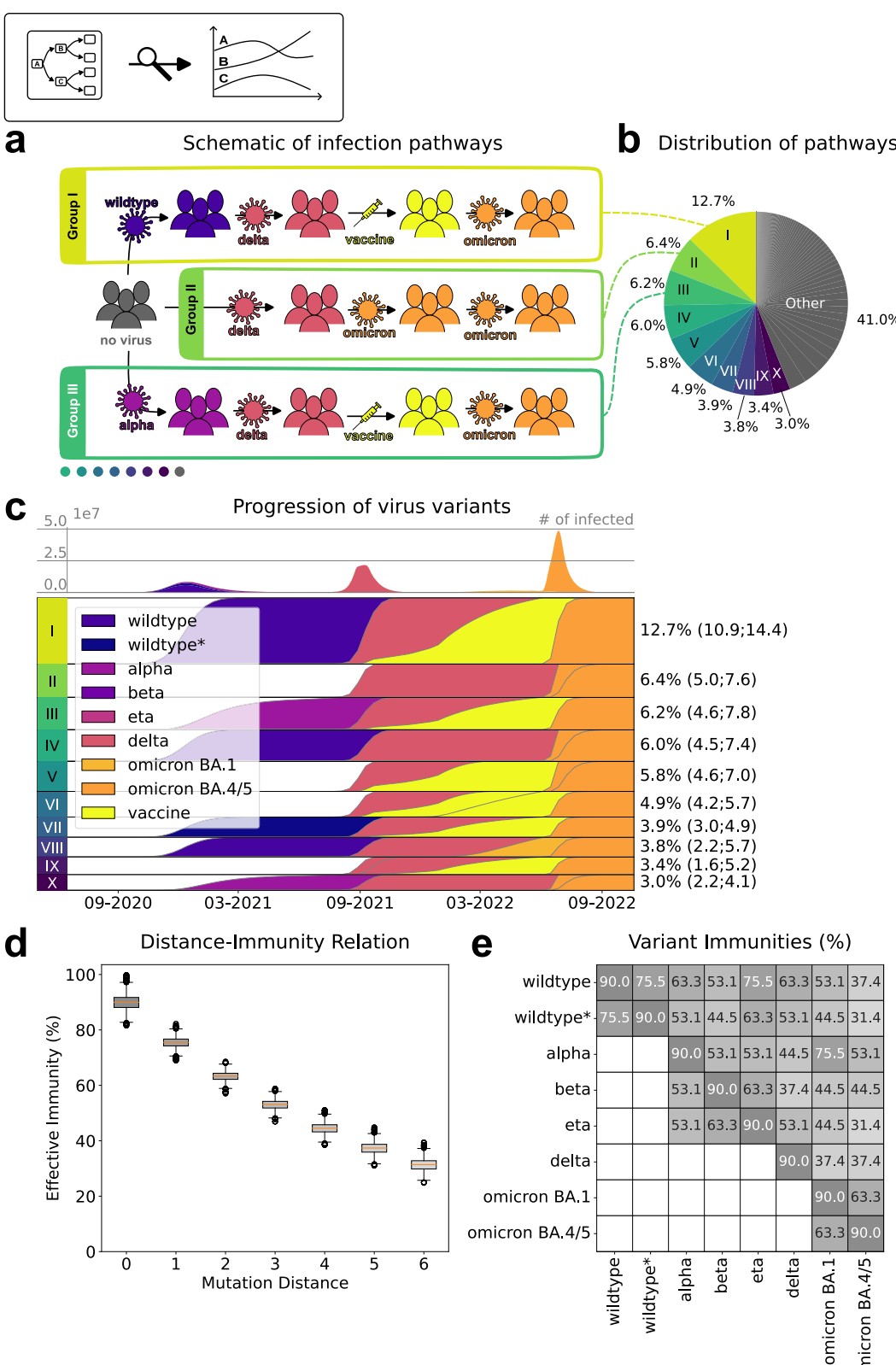

To capture the dynamics of the antibody levels in Addis Ababa and Jimma, the antibody-level model used the available information about variants and vaccinations as inputs. The vaccination rate was calculated as monthly averaged rates based on the vaccination information provided by the participants of the antibody study, and the relative abundance of variants was computed by fitting Gaussian kernels to the data and using them as weights for the time-dependent effective transmission rate, i.e., the weighted sum of all variant transmission rates. The results of these computations can be seen in Fig. SN10 and Fig. SN11 of the Supplementary Information.

Additionally, the model incorporates seasonality, as described for the multivariant model. Furthermore, two immunity factors are introduced as multipliers of the transmission rate: one applied if either

**Fig. 4 | Analysis of estimated variant-pathways and cross-immunities.**
**a** Illustration of three common pathways depicting the progression from a susceptible state to acquiring up to four different infections and/or vaccinations over time. **b** Proportions of variant-pathway-groups within the population, highlighting groups that constitute more than 3% of the total population. **c** Timeline of total number of infected people (first row) and time-resolved compositions of each group highlighting the portions of last variant recovered from or vaccination obtained by color (subsequent rows). **d**–**e** Estimated cross-immunity-levels, with 100% corresponding to a zero percent infection probability and 0% corresponds to infection risk as without previous infection. **d** Boxplot of estimated immunity-levels including sampled uncertainty. Immunity depicted with respect to MOIC mutation distance between newly encountered and previously encountered variants (-combinations) (Center line, median; box limits, upper and lower quartiles; whiskers, 1.5x interquartile range; points, outliers). **e** Heatmap of cross-immunity levels between variants. Y-axis corresponding to previous and x-axis to new variant. Intensity of colors corresponds to strength of cross-immunity, with darker shades indicating higher levels of immunity. Empty cells indicating infection combinations excluded a priori from models based on the world wide variant wave chronology, e.g. a wildtype infection after recovery from delta. **c-e** Median and CIs obtained from $n = 6001$ samples after burn-in from Markov chain Monte Carlo. Source data are provided as a Source Data file.

of the antibody levels is in the *medium* category and a second applied on top of the first factor if either level is in the *high* category.

Unlike for the multivariate model, the distribution of variants is derived a priori from available data, and only their transmission rates and initial time of the overall disease dynamics are estimated. Incubation and recovery times are also estimated. For further information on the model setup, parameter details, and estimation results, we refer readers to the supplementary materials.

The antibody-level model possesses several unknown parameters, including the rates of antibody waning, the infection rates for different variants, and the fraction of infections, which directly result in a *high* Anti-N category. We estimate these parameters using Markov chain Monte Carlo sampling from the available data, which are the fraction of individuals in different categories and the national PCR test positivity rate. The parameter estimation provided a model which describes all these data well (Fig. 5b and c (left)). Indeed, credible intervals for parameter estimates (Supplementary Information Table SN8), state variables (Fig. 5b) and predictions (Fig. 5c) were mostly tight, indicating a low uncertainty of model predictions. In alignment with immune escape properties of later variants, we estimated higher valued infectiousness parameters for them, e.g. omicron BA.4/5 having 3.3 times the delta and 10.6 times the wildtype infectiousness. Relative infectiousness for all variants can be deduced from Supplementary Note Tables SN3 and SN6.

As for the multivariant model there is some discrepancy between national test positivity rate and model description (which might be caused by different regions in Ethiopia). Nevertheless, the antibody-level model provides an accurate description of the available antibody data, so that we used it to predict the current antibody levels, including observations of antibody levels until April 2022. We found that following the omicron wave, our model predicts a remarkable trend (Fig. 5b): up to 100% of the population is projected to fall into the high antibody category for both Anti-N and Anti-S antibodies. This prediction is subject to minimal uncertainties. Notably, the parameter estimation determined slow decay of both Anti-N and Anti-S antibody levels, leading to sustained high levels in the high antibody category until present times.

Given that the sequence of infections and vaccinations was predicted to yield high antibody levels, we explored the impact of vaccination rates. In addition to the actual reported vaccination rate, we considered a 5- and 10-times increased vaccination rate (Fig. 5c, middle), two levels, which could have been achieved using redistribution on the global scale. The artificial experiments indicated that increased vaccination rates would have led to a substantial reduction in infections during the delta wave. For the omicron wave, a reduced impact is predicted due to the higher transmission rate, but the number of hospitalizations could have been substantially lower with higher vaccination rates.

The second type of prediction involved retrospectively examining the impact of varying vaccination rates on the overall virus spread. By multiplying the actual vaccination rate by different factors larger than 1, we investigated how improved vaccination scenarios could have affected the course of the pandemic. Our analysis reveals compelling insights: a vaccination rate five times as high as the actual rate, equivalent to 11.2e7 vaccinated dosages instead of the actually observed 2.7e7, would have significantly mitigated the delta wave. Furthermore, higher vaccination rates of 5 or even 10 times the actual rate could have substantially reduced infections during the omicron wave, potentially halving or lowering it even further (Fig. 5c second and third subplot).

Overall, the predictions of the antibody-level model highlight the critical role of early vaccination in controlling the spread of the virus and provide valuable information for policymakers and public health officials. The results of our model offers evidence-based projections that shed light on the potential outcomes of different vaccination scenarios, emphasizing the importance of accelerated vaccination efforts early on in curbing the impact of viral variants, while implying a minor role of later vaccinations in already saturated natural immunity level scenarios.

## Discussion

The course of the COVID-19 pandemic and current immunity status for many countries is still not sufficiently understood to inform decision-making about the effectiveness of past measures and strategies for future pandemics. This study provides data and model-based analysis to close some of the gaps for Ethiopia. By performing wide sampling before the omicron wave and quantifying antibody titres, we provide insights into the cumulative infection numbers, including the prevalence of reinfections. This suggests that by the end of the last sampling round in April 2022, already 55.1% of the inhabitants of Ethiopia recovered from two SARS-CoV-2 infections. Another 4.1% of the inhabitants of Ethiopia recovered from three SARS-CoV-2 infections. Comparing this to the roughly 470,000 officially confirmed case numbers at the end of April 2022 and the official WHO number of 500,000 cases by late spring 2023 (WHO Covid-19 Dashboard), it is clear that drastic underreporting regarding the number of SARS-CoV-2 infections has been and is still happening in Ethiopia.

Our broad longitudinal analysis of PCR-positive swabs complemented the information about antibody levels and provided an overview of disease-driving mutations. In Ethiopia, wildtype, alpha, delta and Omicron BA.4/5 were the most influential SAR-CoV-2 variants and appeared (except alpha) with a slight delay compared to the global appearance (Supplementary Information Fig. SN9). In relation with the Ethiopian variant survey of Sisay et al. our key findings are confirmed[9]: The importance of B.1.480 (wildtype* in our case) and non-concerning B.1 sublineages (wildtype in our case), the minor role of beta and the general timeframe and dominance of alpha, delta, and omicron waves are common discoveries. Since the observation period of Sisay et al. ends in February 2022, which is around the time when the statistical power of our sequencing data decreases substantially, future research about the precise transition between the omicron waves BA.1 and BA.4/5 could be worth exploring.

To fully exploit the large datasets, we developed two models in this study. The multivariant model provides, to the best of our knowledge, one of the most detailed descriptions of the dynamics of the SARS-CoV-2 pandemic for an African country and is unique as it

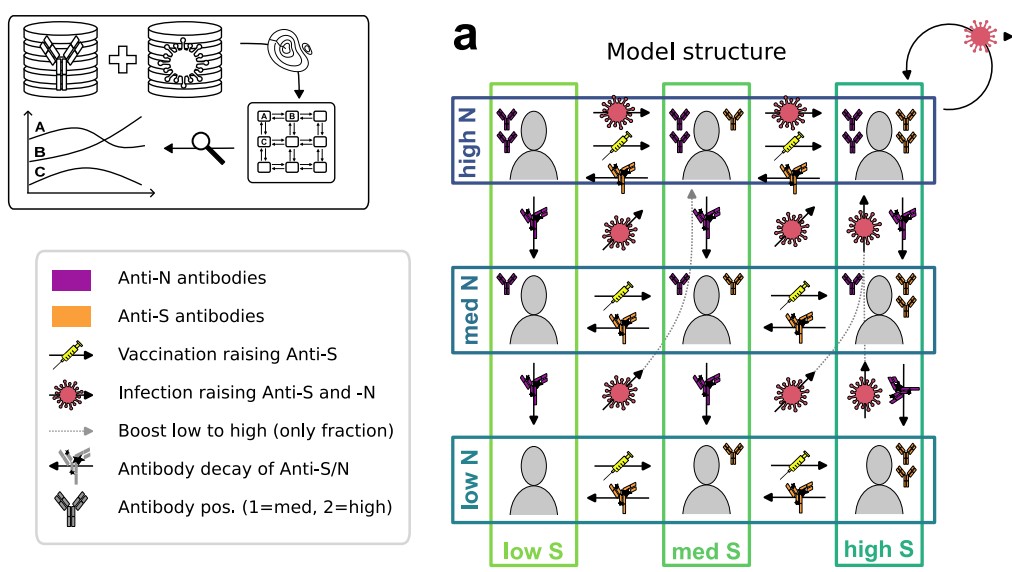

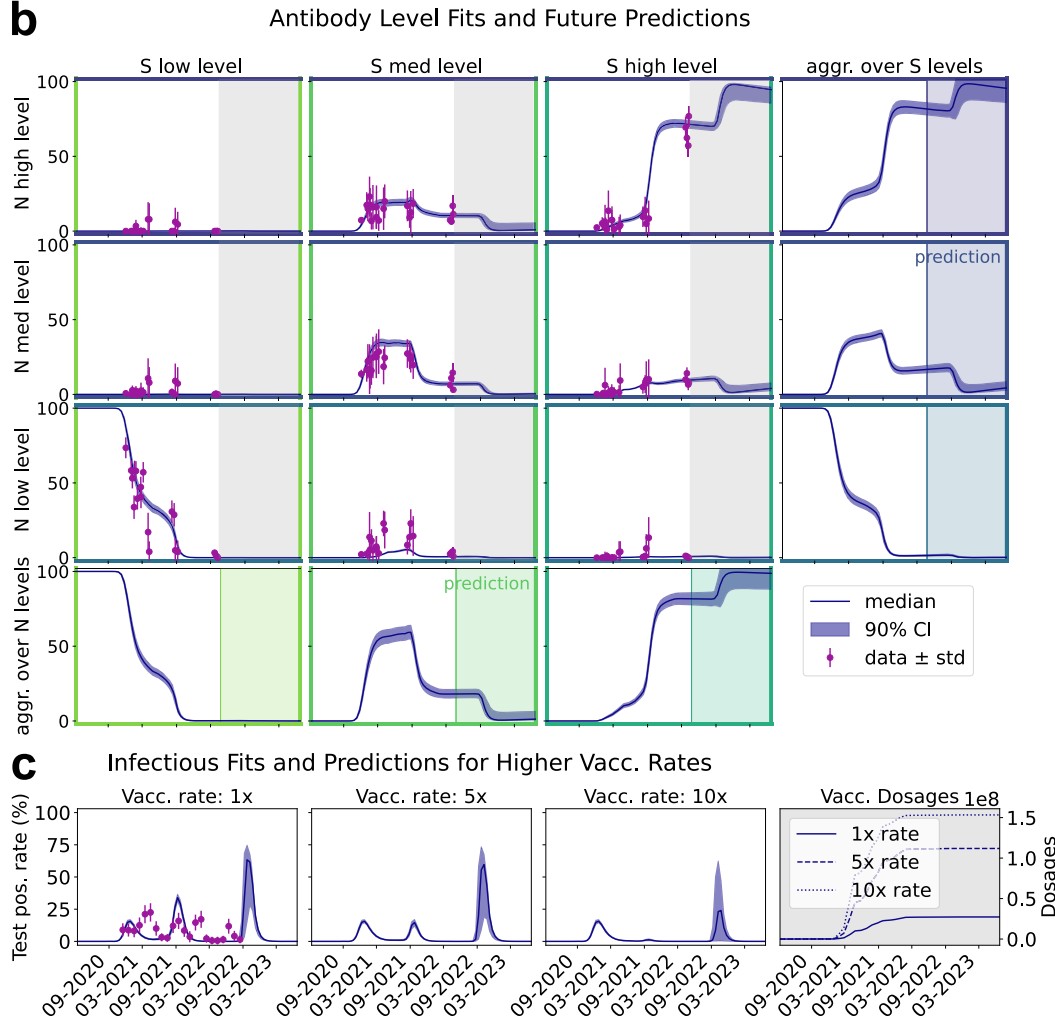

**Fig. 5 | Modeling, fitting results and predictions of antibody-category model.**
**a** Model structure of different antibody category levels and transmission between them. Antibody levels against N or S, respectively, are indicated by colors. **b** The inner grid presents the fitting results to mean measurements (with standard deviations) along with 90% confidence bands derived from parameter sampling. The aggregated levels of Anti-S and Anti-N antibodies are displayed in the lower row and right column, respectively. The prediction phase, where no new antibody data was measured, is highlighted. **c** First plot showing the fit of simulated incidences to

measured national test positivity rates (mean and standard deviation taken per month). Second and third plots illustrating predictions of test positivity rates under hypothetical scenarios with vaccination rates 5 and 10 times as high as the actual rate. Last plot showing how different vaccination rates translate to vaccinated dosages. **b, c** Prediction simulations performed on *n* = 30,001 parameter samples after burn-in from Markov chain Monte Carlo. Sample sizes of data points provided in Supplementary Note 3 (Table SN7). Source data are provided as a Source Data file.

allows for the description of multiple waves and the variant replacement dynamics. Many of the important studies on African countries presented so far focus on individual waves[19–21], a specific variant replacement event[22,23], or do not explicitly account for variants[24,25]. Here, we showed that our model provides a new way to assess pathways of infections and vaccinations as well as cross-immunity between variants with low prediction uncertainties. By integrating three complementary datasets: antibody, variant, and test positivity data, the model identified the four most dynamic driving variants and accurately mapped the timing of large-scale occurrences of second infections to the delta wave. Surprisingly, with the omicron variant, almost the entire population had a second infection, and third infections also became relevant. The investigation of cross-immunity revealed that a simple model based on the distance in MOIC is sufficient to describe the observed data. The model predicts cross-immunities ranging from 24.5% to 68.6% risk reduction.

The estimates and predictions provide an in-depth assessment of the situation in Ethiopia. On the high level, they also agree with other studies, including the meta-analysis by the COVID-19 Forecasting Team, which used Bayesian meta-regression to pool results of 65 studies from 19 different countries on protection against new variants by past infections with earlier variants[26]. For pooled protection against ancestral variants, which the COVID-19 Forecasting Team uses as a collective term for all variants which occurred earlier than the alpha variant, they obtained protection levels of 84.9% (72.8, 91.8). Comparing their result (95% CI) to our findings (with 90% CIs) of 90.0% (85.3, 94.9) of wildtype and wildtype*, which in our context corresponds to variants earlier than alpha variant, against themselves and 75.5% (72.2, 78.7) against each other, we see that our estimates lay well inside the study's CI (Fig. 4e, Supplementary Fig. 2). Pooled protection against the alpha variant is stated to be 90.0% (54.8, 98.4) while our values range from 53.1% (50.2, 56.0) to 90.0% (85.3, 94.9) (Fig. 4e, column on alpha variant), where our lowest median is only slightly below their CI's lower bound and the CIs overlap (Supplementary Fig. 2). Protection against beta is reported to be 85.7% (83.4, 87.7). Since the beta variant did not play a large role in Ethiopia according to our data, it is not surprising that this very tight interval is not represented by the values of our beta column in Fig. 4e. The eta variant was not explicitly investigated by the COVID-19 Forecasting Team. Delta induces reported protection of 82.0% (63.5, 91.9). Our model suggests lower protection values despite that wildtype, which is the main variant after which delta reinfections happened according to our model, has a median of 63.3% (60.6, 65.9), i.e. for delta the CIs are overlapping. For omicron BA.1, the COVID-19 Forecasting Team states protection levels of 45.3% (17.3, 76.1), which completely covers our values for previous infection with other variants. Only reinfection with BA.1, which does not play a role in our findings, is above this interval. For a meta-analysis on BA.4/5, there were insufficient publications available. They only cite one study[27] with protection levels of 76.2% (66.4, 83.1) for previous omicron BA.1 and 35.5% (12.1, 52.7), where the former is only slightly undercut and the latter slightly exceeded by our median values depicted in the last column of Fig. 4e. Overall, for the variant-variant combinations, which play a major role according to our model and are also part of the meta-study, the cross-immunities we obtained are mostly in accordance with the COVID-19 Forecasting Team's findings. The other variants must be treated more cautiously since either their minor role in our model makes it difficult to compare to the pooled data of the meta-study or the meta-study lacked sufficient statistical power to report on them.

The analysis based on the multivariant model was complemented using a tailored model for the description of antibody levels. The analysis of the available data using this model suggested that antibody decay is slow, in particular for Anti-S antibodies. This is in accordance with other research on SARS-CoV-2 antibody decay[28], although direct comparison of numbers is difficult due to the 2-dimensionality and 3-category setup of our model, tackling the issue of limited individuals participating in all rounds of data collection. Van Elslande et al. reported a median time to 50% seronegativity of 809.6 days in non-severe patients (resp. 985.9 days for severe cases) for Anti-S and 273.1 days in nonsevere patients (resp. 327.3 days for severe cases) for Anti-N[28]. The decay is assay-specific and thus, should be interpreted based on the test system used. We have investigated the decay in unpublished longitudinal cohorts in Munich using the same test system as this study (Ro-N-Ig and Ro-RBD-Ig-quant, for details see methods section) and see similarly slow decay of Anti-N and even slower decay of Anti-S signals. In accordance with the results, our antibody model indicated that, particularly with respect to the S-protein, antibody levels remain in the *high* category in the population to date, suggesting that current vaccinations may have a negligible effect. This is based on the general population, and thus does not take into account additional needs of vulnerable groups which might still benefit from vaccination in this setting of recurrent infection waves. Furthermore, by simulating higher vaccination rates retrospectively, we concluded that it would have been possible to substantially mitigate the delta and omicron waves with more administered vaccines. For the delta wave this is strongly supported by our healthcare worker antibody data, where in August 2021 most of the *high* antibody levels were caused by vaccination in comparison to community members with almost no vaccination, but similarly *high-level* percentages (Fig. 1d, Supplementary Fig. 3d). On the other hand, for omicron we have high uncertainties in our predictions (Fig. 5c). Taking into account the high immune escape property of omicron we would probably still have seen a substantial wave, nevertheless with a notably smaller peak. Moreover, from then on most of the population was exposed multiple times and thus benefits of the titres are less pronounced now.

It is important to approach these findings with caution, since we assessed total levels of antibodies, not neutralizing levels, and the relationship between overall antibody levels and reinfection risk is still an area of ongoing research. There is literature confirming that relative reinfection risk after first infection is around the median 32% that our multivariant model estimated. For example, Iversen et al. present 35% relative risk after first infection of Danish healthcare workers[29]. Transfer of protection data from the literature to Ethiopia is complicated, as the conditions of most studies in the field are vastly different. Protection varies considerably depending on the width of the pre-existing immune response and the time between last exposure and the exposure in question. The magnitude of the measured antibody levels also varies depending on the specificity profile of the antibodies and antigens used in the tests. With larger differences in antigenic structure, cross-protection decreases and variation in the serology results increases.

We focused on analyzing data from community members to investigate the antibody progression associated with SARS-CoV-2 infection. Virus variant-specific information was available for isolates from the clinics also derived primarily from community members and not specifically for healthcare workers. A detailed analysis of the antibody progression among healthcare workers can be found in Supplementary Fig. 3.

The study presented here provides several new insights, but also has weaknesses. On the data collection side, the low number of sequenced swabs after the end of 2021 is problematic. We thus accounted for inhomogeneous sampling in the statistical analysis and the parameter estimation. The models we propose here are based on antibody and variant data from Addis Ababa and Jimma, as well as nation-wide test-positivity rates. While the sampling regions in Addis Ababa and Jimma cover areas of different population density and should prove a broad picture, they might not be fully representative for the spread of SARS-CoV-2 in Ethiopia. An indication for this is that the nation-wide test-positivity rate increases in April 2021 and January 2022, while the antibody data do not show substantial changes at these

time points or briefly afterwards. Hence, the use of the combined dataset for the assessment of Ethiopia is an extrapolation. Moreover, (i) the description of cross-immunity factors as a function solely depending on MOIC neglects that other mutations might also affect immune escape potential, (ii) the dependency of cross-immunity after infections with different variants on the union of mutations from previous variants might overemphasize later variants (since secondary infections are assumed to mainly recall cross-reactive antibodies). Yet, these simplifications were important to ensure computational feasibility and balance model complexity and statistical power in the data. A consideration of all mutations would have increased the number of model parameters by a factor of 9.5 and the dataset would have been insufficient to inform them. Despite its limitations, this study provides an unprecedented insight into the dynamics of COVID-19 infections over time and the impact of the variants in Ethiopia. The findings have valuable implications for current and future research and policy-making, enabling a better understanding of the actual situation and offering potential directions for vaccination policies.

To conclude the dynamics of the SARS-CoV-2 variants in Ethiopia between 2020 and 2022 had similar trends as those observed globally. However, our five rounds of seroepidemiological survey in Addis Ababa and Jimma between August 2020 and April 2022, revealed that in our study group over 96% were exposed at least once to the virus by the last round of our survey. This figure is much higher than in other nation-wide reports. Combining longitudinal serology, viral sequencing data, national test positivity rates, and mathematical modeling, we conclude that most Ethiopians have had multiple exposures to SARS-CoV-2, leading to high antibody titres with slow decay characteristics. Due to recurrent infections with different variants and vaccination in many individuals in Ethiopia, we expect a strong hybrid immunity to date.

The models developed based on the antibody and virus variant dynamics show that earlier and more widespread vaccination of the population would have reduced the overall number of infections considerably. However, the general population has now undergone multiple infections as detected by serology and most likely will not benefit much from further vaccinations, especially if the vaccine still harbors the wild type receptor binding domain sequences. Due to persistent circulation of the virus with obvious underreporting, the main focus for preventive actions should be focused on the most vulnerable groups of the population.

## Methods
### Ethics
In this study, samples were collected as a follow-up to our previously published work[1].

In brief, we conducted a follow-up investigation on antibody prevalence at two centers in Ethiopia: Jimma Medical Center [JMC] in Jimma and St Paul's Hospital Millennium Medical College in Addis Ababa. The research was approved by the Institutional Review Boards of Jimma University Institute of Health (IHRPGD/978/2020 and IHRPGD/361/2021) and St Paul's Hospital Millennium Medical College (PM23/239/2020 and PM23/003/2020) as well as Ludwig Maximilian University of Munich (21-0293). Further approval from Addis Ababa and Oromia Regional Health Bureaus was also obtained (BEFO/KBTFU/1-16/488). Written informed consent in local languages was obtained prior to admission to the study. For participants unable to read or write, an impartial witness was involved and fingerprints were obtained for consent. Preliminary results were presented to the Ethiopian Public Health Institute, Federal Ministry of Health of Ethiopia, and Ethiopian Medical Association.

### Antibody data acquisition
Community members and healthcare workers were recruited for the serology study based on convenience sampling. Hospital workers – including clinical staff, medical interns, cleaners, guards, food handlers, and administrative personnel – were recruited at two hospitals, the St Paul's Hospital in Addis Ababa and the Jimma Medical Center in Jimma. In Addis Ababa, community members from Addis Ketema and Yeka subcities were recruited. In Jimma, no specific region was chosen and rural participants were recruited around the Jimma Zone. Sample sizes were initially calculated in July, 2020, when not much baseline data was available and later became flexible as more data became available. Moreover, as the rate of dropout was more than 30% (our initial expectation), we recruited more participants to compensate for the dropouts (c.f. Supplementary Fig. 1 for detailed studyflow). One participant per household was sampled to avoid any clustering effects and households were selected randomly in a way that avoided frequent interaction from the next candidate household to prevent cross-contamination. Overall the median age was 30 with 90% percentile (20,60) and 55.6% of participants, which provided information about sex were female (for round and site-specific demographics see Table 1 and Supplementary Table 1). All participants of the first 3 rounds were enrolled before the introduction of COVID-19 vaccines in Ethiopia. In later rounds participants provided their vaccination status and dates through a questionnaire. For more details see in-depth description in Gudina et al.[1].

In total, 3 ml of venous blood was collected in standard serum tubes. After full coagulation at room temperature, serum was harvested by centrifugation and stored at −20 °C on the same day as sampling. The Roche Elecsys® anti-SARS-CoV-2 [Ro-N-Ig] and the Roche Elecsys® anti-SARS-CoV-2 S [Ro-RBD-Ig-quant] were used for serologic analysis. Both assays are double-antigen sandwich assays, detecting antibodies of all subclasses against SARS-CoV-2. Measurements were performed on a Cobas e801 analytical unit (Roche Diagnostics, Basel, Switzerland) in Munich, Germany, or a Cobas e601 unit (Roche Diagnostics, Basel, Switzerland) in Jimma and Addis Ababa, Ethiopia, using electrochemiluminescence (ELECSYS) technology.

The Ro-RBD-Ig-quant assay uses a truncated S1 protein as an antigen and is a quantitative assay validated for use with human serum and plasma. It is linear between 0.4 and 250 Units (U) per ml, which are equivalent to the standardized (WHO publication WHO/BS.2020.2403) BAU (Binding Antibody Units) according to the manufacturer's manual. Values above 250 U/ml were diluted in 10-fold until the linear range was reached according to the manufacturer's procedures. Values in this study were measured within the linear ranges and back-calculated depending on the dilution as appropriate.

The Ro-N-Ig assay is a qualitative assay similar to Ro-RBD-Ig-quant, but using nucleocapsid as an antigen. The results are given as cut off index (COI), and only a cutoff for positivity is provided by the manufacturer. A linear range is not officially established. We use the raw COI values in a semi-quantitative manner, as we have observed a good dynamic range and excellent repeatability of the values. Anti-N measurements were not diluted, so can be outside the linear range in this work.

### Variant data acquisition
A total of 1873 SARS-CoV-2 RT-PCR positive swabs were collected in Jimma and Addis Ababa, Ethiopia between October 2020 and July 2022. Sample dates were not always available as an exact date, but rather month and year only. Therefore, the midpoint of the respective sampling month was used for all samples analyzed. The swabs were collected from individuals presenting with COVID-19-related symptoms, contacts of confirmed COVID-19 cases, and high-risk populations such as healthcare workers. The specimens were collected at Jimma Medical Center and St. Paul's Hospital.

Jimma Medical Center in Jimma Town and St. Paul Hospital in Addis Ababa are among the major COVID-19 testing and treatment sites in Ethiopia. Jimma COVID-19 center serves as the only COVID-19 diagnostic facility in southwest Ethiopia, home to about 20 million

inhabitants. It is the only facility with intensive care for severe COVID-19 cases in the region. St. Paul Hospital in Addis Ababa is a public tertiary referral hospital serving as a COVID-19 diagnostic and treatment center for Addis Ababa and surrounding areas.

All RT-PCR-positive specimens were stored at −80 °C at these two sites during the study period. Specimens in poor storage conditions and those without proper documentation of data collection dates were excluded. The stored samples were transported on dry ice to Munich in Germany. There, whole nucleic acid extraction was performed using the tanbead maelstrom 4800 instrument (TANBead, Taiwan) and the TANBead Optipure Viral Auto Tube / Plate extraction kits (TANBead, Taiwan). cDNA of the extracts was generated using the LunaScript one step RT (New England Biolabs).

Following the ARTIC network nCoV-2019 sequencing protocol v2[30], amplicons spanning the whole SARS-CoV2 genome were amplified from the cDNA samples. The resulting products were pooled, tagmented with NexteraXT library prep kit (Illumina, San Diego, USA), barcoded, and sequenced on an Illumina NextSeq 2000. For each sample, the sequenced reads were demultiplexed and mapped to the SARS-CoV-2 reference genome (NC 045512.2) with bwa-mem[31]. The consensus sequences were obtained from the sequenced amplicons using the iVar package[32]. Briefly, the package trims the primer sequences from the mapped reads and filters them by a base quality >20 and minimal read length of 30 nt. Pileup files are generated from the mapped reads which are used to assemble the consensus sequence. The consensus sequence was assigned to SARS-CoV-2 lineages using the Pangolin tool[33].

## Analysis of antibody data

To ensure a broad analysis, we merged the data collected for community members in Addis Ababa and Jimma for each round, as the timing of the sampling campaigns overlapped significantly. This allowed us to combine the data effectively and to capture a more comprehensive picture of the antibody dynamics in these communities.

To facilitate meaningful analysis while preserving the relative order of magnitude and accounting for zero measurements, we transformed the antibody measurements using the shifted logarithm base 10 function ($\log_{10}(x+1)$). This transformation enabled us to easily analyze the data across different scales while still maintaining the interpretation of zero as the absence of detectable antibodies.

For categorizing the antibody levels, we considered measurements for each antibody type independently, disregarding the round in which they were obtained. Anti-N values and measurements below the predefined cutoff were excluded from the analysis. We performed k-means clustering with two means, i.e. $k = 2$, on the remaining samples to assign the measurements into distinct antibody level categories.

Smooth changes in antibody levels over time were visualized using a monotonic spline-fitting approach. This allowed us to capture the overall trend and highlight gradual variations in the antibody responses.

To ensure an adequate number of data points for model fitting while remaining reasonable errors for the analysis, we performed k-means clustering ($k = 2$) on the dates of each round. Subsequently, we split each round into two subgroups based on the clustering results and aggregated the antibody responses within these subgroups. Additionally, to estimate high-confidence intervals for error analysis, we fitted a multinomial model to the distribution of the three antibody categories.

To estimate vaccination rates in our study, we employed a fitting approach using monotonic splines applied to the vaccination information provided by the participants, allowing us to capture the temporal trends and variations in vaccination rates accurately. For comprehensive details on the specific methodologies and results of the vaccination rate estimation, we refer readers to the supplementary materials.

For more detailed information and results, we encourage readers to refer to the corresponding sections in our manuscript.

## Analysis of variant data

Whole genome sequencing and subsequent analysis utilized Nextstrain's[34] Augur software, coupled with Auspice for phylogenetic analysis and visualization. To classify the sequenced genomes, we employed pango lineages[33] and grouped them based on shared mutations of interest or concern (MOIC) on the S1 protein according to outbreak.info[12,13]. To quantify the genetic distances between these variant groups, we utilized the Hamming distance, a metric often used to measure distance in gene alignment[14–16]. Here we calculate the distance only based on different MOIC and not all mutations to grasp only the major immune escape changing differences. For our models below we allow for additional behavioral differences independently of this distance. To capture the temporal dynamics of variant prevalence, we organized the samples according to the month of collection and calculated the fractions of each variant. For a smooth visualization of these trends, we applied monotonic spline fitting to generate smoothed curves. To estimate the errors for later parameter estimation, we utilized a multinomial model and fitted it to the monthly variant distributions. To obtain an input function of variant distribution for the antibody level category model while maintaining a reasonable level of complexity, we aggregated the samples over two-month intervals before applying monotonic spline fitting. These procedures allowed us to effectively characterize the variant dynamics and obtain essential inputs for subsequent modeling analyses.

## Modeling

The model-based analysis was performed using compartment models. Utilizing the SEIR (susceptible, exposed, infectious, and recovered) framework, which has been shown to be reliable for modeling the spread of Covid-19[1,35], we aimed to analyze and predict the dynamics of the pandemic.

For the multivariant model we constructed pathways, i.e., chains of SEIR strands, allowing up to four consecutive infections or vaccinations. Pathways which deviated from the chronological order of variant appearances worldwide were excluded. Furthermore, the model only allows for a third infection with the two omicron variants and a fourth infection exclusively by omicron BA.4/5 to account for the reported inter-infection intervals. We allow for different transmission rates for each variant—thereby implicitly considering all mutations – and model their cross-immunity as a function of difference in MOIC. Rates for first, second and third vaccination were estimated a priori as splines from the vaccination information of the antibody study participants and implemented as time-dependent functions into the model.

The antibody-level model does not trace pathways of variants and infections, but categories of antibody levels for Anti-S and Anti-N. Here the SEIR strands are connecting the categories allowing for a boost in antibody levels by infection and recovery. For this model the vaccination is calculated a priori as an average vaccination rate and implemented as a time-dependent function into the model. Moreover, we made the assumptions that people with already high Anti-S levels do not get vaccinated anymore, i.e., the amount of people still applying for vaccination after two infections or vaccinations is negligible. Because of the non-pathway nature of this model we also fitted the variant distribution a priori and used this fit as weights for a sum over the variants' transmission rates to obtain an effective transmission rate. The exact formula for this can be found in Supplementary Note 3 of the Supplementary Information.

The models were encoded using the Systems Biology Markup Language (SBML)[36] and simulated via the software toolbox AMICI[37]. More comprehensive details regarding the modeling methodology are provided in the model subsections of Supplementary Notes 2 and 3 of the Supplementary Information.

## Parameter estimation

To estimate the model parameters, we adopted a Bayesian approach, integrating categorial antibody data and sequenced variant information, along with national test positivity rates and previous knowledge derived from the literature regarding disease progression rates. The model parameter inference was performed using an adaptive Metropolis-Hastings algorithm from a starting point estimated with frequentistic, gradient-based optimization, both expertly implemented in the Python Parameter Estimation Toolbox (pyPESTO)[38]. In order to capture the temporal dynamics of the antibody levels, we split each antibody round into early and late phases using the k-means clustering technique. The resulting samples from the posterior distribution were post-processed, e.g., by removing the burn-in, and convergence was assessed visually and using the Geweke test. The samples were then utilized to derive predictions and associated credible intervals (CIs), providing valuable insights into the dynamics of the pandemic. The parameter estimation problems were formulated using the Parameter Estimation table (PEtab)[39] standard. More information on the parameter estimation setup and results can be found in the corresponding subsections of Supplementary Notes 2 and 3.

## Reporting summary

Further information on research design is available in the Nature Portfolio Reporting Summary linked to this article.

# Data availability

The models and population average data are available at Zenodo [https://doi.org/10.5281/zenodo.10871139]. The variant sequences are published in the Sequence Read Archive[40] under project number PRJNA1017685. Individual level data will be made available to other researchers in a reasonable timeframe upon qualified request to the corresponding authors AK and AW, due to limitations of data sharing in the ethics statements. Source data are provided with this paper.

# Code availability

The code for model creation,data aggregation and figure plotting is available at Zenodo [https://doi.org/10.5281/zenodo.10871139].

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

## Acknowledgements

We are grateful for research funding provided by the Bavarian State Ministry of Sciences, Research and the Arts (Bayerisches Staatsministerium, F.4-V0122.4/3/20 (AK, AW)); the Germany Ministry of Education and Research (MoKoCo19; 01KI20271 (JH, AW) and FitMultiCell; 031L0159C (SM, JH) and INSIDe; 031L0297A (SM, JH, AW) and GEN-Immune; 031L0292F (SM, JH)); the EU Horizon 2020 programme (ORCHESTRA; 101016167 (JH, AW)); and Volkswagenstiftung (E2; 99 450 (SM, JH)). This work was supported by the Deutsche Forschungsgemeinschaft (DFG, German Research Foundation) via project funding (SEPAN; 7376/3-1 (JH)), under Germany's Excellence Strategy (project IDs 390685813 - EXC 2047 (JH) and 390873048 - EXC 2151 (JH)), and by the University of Bonn via the Schlegel professorship to JH. This study was partially funded by the Free State of Bavaria under the FORCOVID (MM) and BayVOC (MM) research initiatives. We thank participants, study teams, Jimma Medical Center, Oromia Regional Health Bureau, St Paul's Hospital Millennium Medical College, and Addis Ababa Health Bureau for the support provided during data collection.

## Author contributions

E.K.G., S.A., A.K., J.H., M.H. and A.W. conceived of and designed the study. E.K.G., S.A., W.A., Gize, S.S.B., B.F., D.B., M.G., E.G., D.Y., A.Z. participated in the data and sample collection. S.A., Gize, M.B., R.R.A., I.P., M.G., A.W. performed serologic analysis. M.M., Graf, S.K., and H.B. performed cDNA synthesis and sequencing. K.E., R.K., J.H. and S.M. summarized, cleaned, and analyzed the data. S.M. and J.H. did the modeling and parameter estimation. E.K.G., S.A., A.K., J.H., M.H., M.M., A.W. interpreted the results. S.M., E.K.G., S.A., J.H., A.K., and A.W. drafted the manuscript. All authors contributed to the writing of the final version of the manuscript. S.M., E.K.G., S.A., K.E., and A.W. have accessed and verified the data; all authors accepted responsibility for the decision to submit for publication.

## Funding

## Competing interests

The authors declare the following competing interests: The medical center of the LMU received reagents and an analyzer from Roche with reduced rates for other studies regarding SARS-CoV-2 serology. MH and AW received different consultancy contracts and support for studies regarding SARS-CoV-2 serology, independent of this project. This did, however, not influence the interpretation of the data, or the data reported. The remaining authors declare no competing interests.

## Additional information

[1]Life and Medical Sciences (LIMES), University of Bonn, Bonn, Germany. [2]Saint Paul's Hospital Millennium Medical College, Addis Ababa, Ethiopia. [3]Jimma University Clinical Trial Unit, Jimma University Institute of Health, Jimma, Ethiopia. [4]CIH LMU Center for International Health, LMU Munich, Munich, Germany. [5]Max von Pettenkofer Institute and Gene Center, Virology, National Reference Center for Retroviruses, LMU Munich, Munich, Germany. [6]German Center for Infection Research (DZIF), partner site Munich, Munich, Germany. [7]Laboratory for Functional Genome Analysis, Gene Center, LMU Munich, Munich, Germany. [8]Division of Infectious Diseases and Tropical Medicine, LMU University Hospital, LMU Munich, Munich, Germany. [9]Institute for Medical Information Processing, Biometry and Epidemiology (IBE), Faculty of Medicine, LMU Munich, Munich, Germany. [10]Immunology, Infection and Pandemic Research IIP, Fraunhofer ITMP, Munich, Germany. [11]Unit Global Health, Helmholtz Zentrum München—German Research Center for Environmental Health, Neuherberg, Germany. [12]Institute of Computational Biology, Helmholtz Zentrum München—German Research Center for Environmental Health, Neuherberg, Germany. [13]Center for Mathematics, Technische Universität München, Garching, Germany. [14]Faculty of Medicine, Max Von Pettenkofer Institute, LMU Munich, Munich, Germany. [15]These authors contributed equally: Simon Merkt, Solomon Ali, Esayas Kebede Gudina. [16]These authors jointly supervised this work: Jan Hasenauer, Arne Kroidl, Andreas Wieser. ✉e-mail: jan.hasenauer@uni-bonn.de; akroidl@lrz.uni-muenchen.de; andreas.wieser@lmu.de

