## [Peer Review File · Nature Communications]

Long-term monitoring of SARS-CoV-2 seroprevalence and variants in Ethiopia provides prediction for immunity and cross-immunityREVIEWER COMMENTS

Reviewer #1 (Remarks to the Author):

Merkt et. al. use two models to examine variant dynamics, variant-specific immunity, reinfection patterns, and vaccination effects in Ethiopia. The study leverages both data generated in the study and existing datasets including antibody serology and PCR tests to model variant dynamics and variance in susceptibility to reinfection as function of genetic distances between variants. Using a second model, the authors also show that immunity from early vaccination and infections could have had a positive net effect on the delta and omicron waves resulting in significantly lower cases and deaths. Importantly, cross-immunity between variants ranged from 24% to 69%.

The study is interesting, and methods appear to be well thought. Their findings build on the results from prior studies on the continent and further affirm the gross underreporting of COVID-19 cases and deaths. Although the study is well written and limitations elaborated, I have a few comments for the authors.

Major comments

1. Retrospectively, it suggests that increased early vaccination could have substantially reduced infections during the delta and omicron waves. However, as a large proportion of the population might have already had multiple infections that led to a strong immune response, further vaccination is less likely to have a significant impact now. – From figure one, SN1, it appears to me that in fact, the effect of previous infections is just as strong as, if not stronger than that of vaccination. I think that the statement downplays the role of previous vaccination. In figure SN1, I see that the majority of the population were of unknown vaccination status and round 4 particularly affirms this. For ant-S, there are two peaks of about the same height yet one is largely unknown vaccination status and the other mostly vaccinated. Do the authors assume by default that unknown is equal to vaccinated? The authors also mention in the discussion that nearly 120M infections with 2-3 infections were expected by the last round of sampling, a number larger than that of vaccination by far suggesting that in fact, infections played a more important role in generating of these antibodies.
2. Evolution of antibody levels over time between end of 2020 and April 2022. – Did the authors mean, "accumulation"? as they do not report any results on how the antibodies evolve.
3. Can the authors comment on the suitability of the study area and sample sizes to make nation wide conclusions? Given that they postulate that 96% of the population could have experience 2 or 3 infection by the 5th round.
4. Page 34, line 731. we calculate the distance only based on different MOIC and not all mutations to grasp only the behavior changing differences – I find this rational abit concerning as the authors assume that other mutations not impact the virus in anyway, however, there are many potential MOICs that whose effect has just not been determined or whose role has could be compensatory. I understand that trying to account for all mutations could overly complicate the model but perhaps lumping up all the other mutations and assigning a lower weight to them might help? Importantly, it is necessary for the authors to account for the rest of the mutations.
5. I could not open the zenodo link (10.5281/zenodo.8264102) provided to scripts used to perform the analysis. Additionally, the authors mention that data analysed will only be provided upon request. In the spirit of reproducibility, I encourage the authors to publish both the scripts and data. An episet_id for GISAID or Genbank accession would be sufficient.

Minor comments

None

Reviewer #2 (Remarks to the Author):

General comments

Merkt et al. analyse antibody levels against SARS-CoV-2 and genetic sequencing data from two locations in Ethiopia. These are useful data, particularly as they span much of the pandemic period, and overall the analysis has been well executed.

The antibody data come from serosurveys conducted in two locations (Addis Ababa and Jimma) and in two different populations (healthcare workers and community members). In their primary analysis, the authors focus on community members and combine data from the 2 locations. Focussing on community members seems reasonable, however, it would be useful to also present data separately for each location. It is probable that seroprevalence trends vary between the sites, and an exploration of this variation would be informative.

In addition to the statistical analysis, the study uses the data (antibody and sequencing data) to fit two models of disease transmission. While these models appear reasonable, they do not align well with the PCR positivity data. They fail to capture two prominent peaks in infection and predict a significant peak where none exists. This discrepancy suggests that these model results may have limited predictive value. Emphasizing the weaknesses of these models is essential.

Specific comments

Intro

I.68 "Given our data, serological evidence of past infection in Ethiopia alone suggests more than ten times as many infections by autumn 2022." Ten times as many as the number reported?

I. 80 "leading to adjustments in vaccination policies in various hospitals." More detail needed.

I. 85 "Previous publications touch upon this topic hypothetically". More detail needed.

I.109 "We utilized a second epidemiological model to predict future antibody dynamics, providing insights into the expected long-term immunity landscape in the Ethiopian population, which is critical to prevent massive spread of SARS-CoV-2 and potential health system crisis in the future." It is a strong claim that the model will prevent spread of SARS-CoV-2 and avert a health crisis! Consider rephrasing or deleting this sentence.

Results

Table 1. The high % women (>75%) in Addis Ababa in R1 and R2 should be explained. Also, it would be helpful to add the dates of each survey round to the table (could be as a footnote).

I.136 "Our analysis revealed that in the majority of individuals, Anti-S and Anti-N antibodies were present (Figure 1A–E), suggesting an infection event." Do you mean the % with non-zero values for these 2 assays. What are these %?

Figure 1. This figure is very informative. If I understand correctly, the cut-off values were chosen by applying 1-dimensional k-means separately to anti-N and anti-S antibody levels. Did you try clustering using 2-dimensional k means, i.e. applying k means to anti-N and anti-S simultaneously?

I.144 "Interestingly, most individuals vaccinated also showed reactivity for Anti-N" Please provide the percentage to support this statement.

Figure 3. The model fits the antibody data but it does not fit data on the national PCR positivity rate – it misses two peaks and predicts a significant peak when there is none. This suggests that either the model predictions or the accuracy of the PCR data, or possibly both, should be viewed with caution.

Figure 5. Again the model does predict the PCR positivity rate very well.

Methods

I.621 It would be helpful to provide a brief description of how participants were recruited into the survey rather than just refer to Gudina et al. 2021. The description should include details such as whether it was a convenience sample, the age range of participants, any exclusion criteria, and how data on vaccination status were acquired.

Supplementary Note 2: The description of the model in Supplementary Note 2 could be made clearer with more details. E.g. Does the variant index $i=1, \dots, 8$ correspond to the columns of Table SN2, i.e. 1 = wildtype, 2=wildtype* etc.? Why are the 3rd and 4th infections limited to $k=7, 8$ (omicron?)

Discussion

I.465 "This suggests that a substantial percentage... had already experienced a second or even third infection" Please provide the actual percent of 2nd and 3rd infections?

I.505 "For pooled protection against ancestral variants, they obtained protection levels of 84.9% (72.8, 91.8). Comparing their result (95% CI) to our findings (with 90% CIs) of 90.0% (85.3, 94.9) of wildtype and wildtype* against themselves and 75.5% (72.2, 78.7) against each other," Why are you restricting to wildtype and not provide an estimate of protection against ancestral variants? It appears that you are not comparing like with like.

L.508 "Overall protection against the alpha variant" Please clarify what is mean by "overall protection" – is it the average protection against infection with the alpha variant among those who have previously been infected with any variant?

Reviewer #3 (Remarks to the Author):

- The authors need to specify the type of vaccines being administered in Ethiopia, since certain vaccine platforms may have stronger immunogenicity (mRNA) than others (inactivated/vector based), all of which were used in the region of Africa. It's possible that the stronger immunogenicity of mRNA vaccine with respect to natural infection could also potentially create bi-model pattern of the distribution of the anti-S antibody level, in addition to repeated infections. It would be great if the authors could show the range of antibody level for individuals without prior infection but vaccination only, using the same assay of the study, and how it compare with the anti-S antibody level in this cohort population.
- Line 281-283: The authors use high-level anti-S antibody to infer individuals with at least two infections, based on the bi-model distribution of the anti-S antibody titer distribution as shown in Figure SN1-SN2. However, if we look at Figure SN1, it seems like having vaccination significantly increase the chances of being in the high titer level (the more doses of vaccination, the stronger the bias towards high titer). So it's likely that individuals with only one prior infection with prior vaccination(s) could also end up in the high anti-S antibody level. How would the authors be able to differential repeated infections with hybrid immunity (prior infection and vaccination) purely based on S titer level?
- Line 262: Here the authors consider the protective immunity profile for individuals with more than one prior infections as a simple union of the mutations from the previous variants. However, it's evident now that the ordering of exposures may bias the immune response due to immune memory (imprinting). I.e, secondary infection largely recall cross-reactive antibodies induced by previous variant, with very limited response towards novel epitopes of variant that caused the secondary infection. This shall be discussed as a limitation of the model.

Author's response to reviewers' comments

We thank the reviewers for their constructive comments. Based on their remarks, we have revised the manuscript. We addressed all their comments and tried to resolve the raised issues. Our point-by-point response is provided below.

The teal boxes contain the editor's and the reviewers' comments, while the yellow boxes contain the changes made in the manuscript. To provide some context, the changes (blue) are surrounded by unchanged text (black).

After working through the comments of the reviewers we can now present a thoroughly revised and improved manuscript. In particular we now provide more information on the study setup and vaccinations, made more data available, and discuss limitations, reasoning behind choices and previously unclear technicalities in much more detail and extend.

We hope the reviewers will appreciate this revised version and anticipate their new review reports.

Response to Reviewer #1

Merkt et. al. use two models to examine variant dynamics, variant-specific immunity, reinfection patterns, and vaccination effects in Ethiopia. The study leverages both data generated in the study and existing datasets including antibody serology and PCR tests to model variant dynamics and variance in susceptibility to reinfection as function of genetic distances between variants. Using a second model, the authors also show that immunity from early vaccination and infections could have had a positive net effect on the delta and omicron waves resulting in significantly lower cases and deaths. Importantly, cross-immunity between variants ranged from 24% to 69%.

The study is interesting, and methods appear to be well thought. Their findings build on the results from prior studies on the continent and further affirm the gross underreporting of COVID-19 cases and deaths. Although the study is well written and limitations elaborated, I have a few comments for the authors.

We thank the reviewer for this positive evaluation and address each comment in the following.

1. Retrospectively, it suggests that increased early vaccination could have substantially reduced infections during the delta and omicron waves. However, as a large proportion of the population might have already had multiple infections that led to a strong immune response, further vaccination is less likely to have a significant impact now. - From figure one, SN1, it appears to me that in fact, the effect of previous infections is just as strong as, if not stronger than that of vaccination. I think that the statement downplays the role of previous vaccination. In figure SN1, I see that the majority of the population were of unknown vaccination status and round 4 particularly affirms this. For ant-S, there are two peaks of about the same height yet one is largely unknown vaccination status and the other mostly vaccinated. Do the authors assume by default that unknown is equal to vaccinated? The authors also mention in the discussion that nearly 120M infections with 2-3 infections were expected by the last round of sampling, a number larger than that of vaccination by far suggesting that in fact, infections played a more important role in generating of these antibodies.

We thank the reviewer for pointing out the limited clarity of text and figures. To address it,

we implemented several changes:

1) We changed the labeling “unknown vaccination” in the Figure SN1 to “N/A” (as in not answered), in accordance with the phrasing from Figure 1. This difference is important since there was no vaccine publicly available in Ethiopia up until and including Round 3¹. Because of this information about general vaccine availability in combination with our previous observation that vaccinated individuals are more likely to answer questions on the vaccination status on the questionnaire than unvaccinated individuals, we considered individuals without an answer (“N/A”) as “unvaccinated” for modelling.

2) We provided in the text additional information on the impact of vaccination vs. infection. This is based on the comparison of the observed antibody levels for healthcare workers (Supplementary Figure 1) and community members (Figure 1). In brief, for healthcare workers a clear shift from medium Anti-S to high Anti-S is observed in response to vaccination, but community members reach the same levels by infections alone. Therefore, it seemed reasonable to treat the effect of vaccine on Anti-S levels analogously as the effect of an exposure to the virus (which does not become clear in the previous manuscript version). Hence, we added the above reasoning to the Supplementary Note 1 and included new histograms of vaccination status, making the timeline of vaccination vs infections more clear.

3) We integrate the information on the percentages of individuals with 2 or 3 infections. This information is included in the revised Discussion section to clarify the meaning of “substantial percentage of the 120 million inhabitants of Ethiopia”.

Changes in the manuscript:

Initial submission:

We constructed a multivariant model to investigate the temporal evolution of the SARS-CoV-2 pandemic in Ethiopia. The model accounts for different sequences of infections and vaccination events (Figure 3A). The sequence of infections and vaccinations - to which we refer in the following as pathways - is tracked to determine the immunity status of individuals. Each infection follows the SEIR schematic, with individuals transitioning from being susceptible to exposed, then infected, and finally recovered. The structure of the multivariant model is outlined in Figure 3A using a small number of possible pathways.

...

This suggests that a substantial percentage of the 120 million inhabitants of Ethiopia had already experienced a second or even third infection by the time of the last sampling round in April 2022.

Revised submission:

We constructed a multivariant model to investigate the temporal evolution of the SARS-CoV-2 pandemic in Ethiopia. The model accounts for different sequences of infections and vaccination events (Figure 3A). The sequence of infections and vaccinations - to which we refer in the following as pathways - is tracked to determine the immunity status of individuals. Each infection follows the SEIR schematic, with individuals transitioning from being susceptible to exposed, then infected, and finally recovered. Due to official vaccine availability in Ethiopia only after Round 3¹ in combination with our previous observation that vaccinated individuals are more likely to answer questions on the vaccination status on the questionnaire than unvaccinated individuals, we considered individuals without an answer (“N/A”) as “unvaccinated” for modeling. The structure of the multivariant model is outlined in Figure 3A using a small number of possible pathways.

...

This suggests that by the end of the last sampling round in April 2022, already 55.1% of the inhabitants of Ethiopia recovered from two SARS-CoV-2 infections. Another 4.1% of the inhabitants of Ethiopia recovered from three SARS-CoV-2 infections.

Changes in the supplement:

Initial submission:

-

Revised submission:

There was no vaccine publicly available in Ethiopia until after Round 3¹. Because of this information about general vaccine availability in combination with our previous observation that vaccinated individuals are more likely to answer questions on the vaccination status on the questionnaire than unvaccinated individuals, we considered individuals without an answer (“N/A”) as “unvaccinated” for modelling. This is also supported by official nation wide numbers of people with at least one dose of vaccine, provided by Our World in Data (ourworldindata.org) and depicted in Figure SN3new. Moreover, we treat the effect of vaccine and infection on Anti-S levels analogously. This is based on the comparison of the observed antibody levels for healthcare workers (Supplementary Figure 1) and community members (Figure 1). There from Round 3 to Round 4 for healthcare workers a clear shift from medium Anti-S to high Anti-S is observed in response to vaccination, but community members reach the same levels by infections alone.

Figure SN3new. Histograms of distributions of vaccination information from study participants at each round. “N/A” responses before public availability of vaccine in Ethiopia are highlighted by hatching. For community members official, national vaccination numbers (provided by Our World in Data) are indicated in red above each round and percentages from our data set of “N/A” responses after public availability of vaccines in Ethiopia are displayed inside of the corresponding bars.

2. Evolution of antibody levels over time between end of 2020 and April 2022. - Did the authors mean, “accumulation”? as they do not report any results on how the antibodies evolve.

We meant to use the term “evolution” to refer to the change of levels of antibodies over time and not the antibodies themselves. However, we agree that the wording here can be confusing and changed the manuscript in the following way:

Changes in the manuscript:

Initial submission:

(F-G) Evolution of antibody levels over time between end of 2020 and April 2022.

Revised submission:

(F-G) Antibody levels between end of 2020 and April 2022.

3. Can the authors comment on the suitability of the study area and sample sizes to make nation wide conclusions? Given that they postulate that 96% of the population could have experience 2 or 3 infection by the 5th round.

The community member samples from Addis Ababa were collected in the subcities Addis Ketema and Yeka and for Jimma from Jimma City and four rural districts in Jimma Zone. These districts represent densely and sparsely populated areas as outlined in our previous publication. Moreover, one participant per household was sampled to avoid any clustering effects and households were selected randomly in a way that avoided frequent interaction from the next candidate household to prevent cross-contamination. Additionally, the datasets from Jimma and Addis are similar although the two cities are located quite far apart, further indicating some representativeness. Yet, the estimation of nation-wide conclusions from these restricted sampling areas has obviously limitations. We included more information on the study setup to the manuscript, highlight potential issues with representability in the Discussion section and included the site specific antibody results to Supplementary Note 1. Additionally, we adjusted the statement about 96% of the population having had evidence of exposure to the virus, to clarify our intended meaning.

Changes in the manuscript:

Initial submission:

On the modeling side, the combination of antibody and variant data from Addis Ababa and Jimma with national test positivity rates might be criticized. The latter was done to provide information about the overall number of cases. Despite its limitations, this study provides an unprecedented insight into the dynamics of COVID-19 infections over time and the impact of the variants in Ethiopia.

...

However, the infection dynamics, as evidenced by our five rounds of seroepidemiological survey between August 2020 and April 2022, revealed over 96% of the population had evidence of exposure to the virus by the last round of our survey, a figure not captured by the national report.

...

Community members and healthcare workers were recruited for the serology study (see in-depth description in Gudina et al. 2021²). In Addis Ababa, community members from Addis Ketema and Yeka sub cities were recruited. In Jimma, no specific region was chosen and rural participants were recruited around the Jimma Zone. One participant per household was sampled to avoid any clustering effects.

Revised submission:

The models we propose here are based on antibody and variant data from Addis Ababa and Jimma, as well as nation-wide test-positivity rates. While the sampling regions in Addis Ababa and Jimma cover areas of different population density and should prove a comprehensive picture, they might not be fully representative for the spread of SARS-CoV-2 in Ethiopia. An indication for this is that the nation-wide test-positivity rate increases in April, 2021 and January, 2022, while the antibody data do not show substantial changes at these time points or briefly afterwards. Hence, the use of the combined dataset for the assessment of Ethiopia is an extrapolation. Moreover, (i) the description of cross-immunity factors as a function solely depending on MOIC neglects that other mutations might also affect immune escape potential, (ii) the dependency of cross-immunity after infections with different variants on the union of mutations from previous variants might overemphasize later variants (since secondary infections are assumed to mainly recall cross-reactive antibodies). Yet, these simplifications were important to ensure computational feasibility and balance model complexity and statistical power in the data. A consideration of all mutations would have increased the number of model parameters by a factor of 9.5 and the dataset would have been insufficient to inform them. Despite its limitations, this study provides an unprecedented insight into the dynamics of COVID-19 infections over time and the impact of the variants in Ethiopia.

...

However, our five rounds of seroepidemiological survey in Addis Ababa and Jimma between August 2020 and April 2022, revealed that over 96% were exposed at least once to the virus by the last round of our survey. This figure is much higher than in other nation-wide reports.

...

Changes in the manuscript (continued):

Community members and healthcare workers were recruited for the serology study based on convenience sampling. Hospital workers - including clinical staff, medical interns, cleaners, guards, food handlers, and administrative personnel — were recruited at two hospitals, the St Paul’s Hospital in Addis Ababa and the Jimma Medical Center in Jimma. In Addis Ababa, community members from Addis Ketema and Yeka sub cities were recruited. In Jimma, no specific region was chosen and rural participants were recruited around the Jimma Zone. Sample sizes were initially calculated in July, 2020, when not much baseline data was available and later became flexible as more data became available. Moreover, as the rate of dropout was more than 30% (our initial expectation), we recruited more participants to compensate for the dropouts (c.f. Supplementary Figure 1 for detailed studyflow). One participant per household was sampled to avoid any clustering effects and households were selected randomly in a way that avoided frequent interaction from the next candidate household to prevent cross-contamination. Overall the median age was 30 with 90% percentile (20,60) and 55.6% of participants which provided information about sex were female (for round and site specific demographics see Table 1 and Supplementary Table 1). All participants of the first 3 rounds were enrolled before the introduction of COVID-19 vaccines in Ethiopia. In later rounds participants provided their vaccination status and dates through a questionnaire. For more details see in-depth description in Gudina et al. 2021².

Changes in the supplementary notes:

Initial submission:

-

Revised submission:

Figure SN4new. Antibody data of community members and healthcare workers by site of collection.

4. Page 34, line 731. we calculate the distance only based on different MOIC and not all mutations to grasp only the behavior changing differences - I find this rational a bit concerning as the authors assume that other mutations not impact the virus in anyway, however, there are many potential MOICs that whose effect has just not been determined or whose role has could be compensatory. I understand that trying to account for all mutations could overly complicate the model but perhaps lumping up all the other mutations and assigning a lower weight to them might help? Importantly, it is necessary for the authors to account for the rest of the mutations.

We thank the reviewer for this comment. We see that the formulation of the sentence is not ideal and agree that also mutations besides MOIC can contribute to behavioral changes. Indeed, – and this is why the sentence was so problematic – our model allows for differences in the general transmission rates of the variant groups independently of MOIC through the variant group specific transmission rates β_i . The MOIC are only used to describe the degree of immunity escape, which is a simplifying assumption to limit the number of model parameters

Modelling immunity escape without restricting ourselves to MOIC could in principle be beneficial. Indeed, the first model we developed in the context of this allowed for all-to-all differences by modelling completely on the level of strains. Yet, the number of parameters and state variables grow quickly with number of variants. A model with 8 strains had 205 different immune escape parameters. The resulting fitting parameter estimation problem was ill-posed.

A further argument for the use of MOIC is in our opinion that they are the basis for lineage grouping. We evaluated a hypothetical lineage grouping using all mutations, i.e. taking the lineages listed in Table SN1 of the Supplementary Notes. This resulted in 50 different variants, which would make computationally infeasible and would probably also not be beneficial. For half of these groups the dataset contains only one or two samples, leading to a poor statistical power.

So, in summary, the formulation was misleading and we indeed considered the impact of other mutations by allowing for strain specific transmission rates. An in depth consideration of the impact of all mutations on the immunity escape does not appear possible given the dataset.

In the revised manuscript, we clarified these points and provided additional highlight the flexibility of the model. Moreover we added a section to Supplementary Note 2 discussing alternative model formulations and why they were not considered.

Changes in the manuscript:

Initial submission:

Here we calculate the distance only based on different MOIC and not all mutations to grasp only the behavior changing differences.

...

For the multivariant model we constructed pathways, i.e., chains of SEIR strands, allowing up to four consecutive infections or vaccinations. Pathways which deviated from the chronological order of variant appearances worldwide were excluded. Furthermore, the model only allows for a third infection with the two omicron variants and a fourth infection exclusively by omicron BA.4/5 to account for the reported inter-infection intervals. Rates for first, second and third vaccination were estimated a priori as splines from the vaccination information of the antibody study participants and implemented as time dependent functions into the model.

Revised submission:

Here we calculate the distance only based on different MOIC and not all mutations to grasp only the **major immune escape** changing differences. **For our models below we allow for additional behavioral differences independently of this distance.**

...

For the multivariant model we constructed pathways, i.e., chains of SEIR strands, allowing up to four consecutive infections or vaccinations. Pathways which deviated from the chronological order of variant appearances worldwide were excluded. Furthermore, the model only allows for a third infection with the two omicron variants and a fourth infection exclusively by omicron BA.4/5 to account for the reported inter-infection intervals. **We allow for different transmission rates for each variant – thereby implicitly considering all mutations – and model their cross-immunity as a function of difference in MOIC.** Rates for first, second and third vaccination were estimated a priori as splines from the vaccination information of the antibody study participants and implemented as time dependent functions into the model.

Changes in the manuscript (continued):

Initial submission:

On the modeling side, the combination of antibody and variant data from Addis Ababa and Jimma with national test positivity rates might be criticized. The latter was done to provide information about the overall number of cases. Despite its limitations, this study provides an unprecedented insight into the dynamics of COVID-19 infections over time and the impact of the variants in Ethiopia.

Revised submission:

The models we propose here are based on antibody and variant data from Addis Ababa and Jimma, as well as nation-wide test-positivity rates. While the sampling regions in Addis Ababa and Jimma cover areas of different population density and should prove a comprehensive picture, they might not be fully representative for the spread of SARS-CoV-2 in Ethiopia. An indication for this is that the nation-wide test-positivity rate increases in April, 2021 and January, 2022, while the antibody data do not show substantial changes at these time points or briefly afterwards. Hence, the use of the combined dataset for the assessment of Ethiopia is an extrapolation. Moreover, (i) the description of cross-immunity factors as a function solely depending on MOIC neglects that other mutations might also affect immune escape potential, (ii) the dependency of cross-immunity after infections with different variants on the union of mutations from previous variants might overemphasize later variants (since secondary infections are assumed to mainly recall cross-reactive antibodies). Yet, these simplifications were important to ensure computational feasibility and balance model complexity and statistical power in the data. A consideration of all mutations would have increased the number of model parameters by a factor of 9.5 and the dataset would have been insufficient to inform them. Despite its limitations, this study provides an unprecedented insight into the dynamics of COVID-19 infections over time and the impact of the variants in Ethiopia.

Changes in the supplement:

Initial submission:

-

Revised submission:

Alternative Model Formulations

Initially we considered three potential model extensions: (i) Describing cross-immunities independently of MOIC. (ii) Allowing all pathways between variants. (iii) No grouping of variants. In the end all of these formulations proved impractical. For (i) we would have to model individual parameters for each combination of past infections and new infections. Even with the other simplifications of the model still in place this leads to a total of 205 immune escape factors instead of the two we have in the current model. For such a high dimensional parameter estimation the dataset would have been insufficient to inform. (ii) would result in a model with 12289 different states being computationally infeasible. Extension (iii) implies 50 different variants instead of the current 8 lineages. Even if we disregard the low statistical power we have for some of these single sublineages, we would still end up with more than 10000 different model states and five times as many parameters as in our current model, make this computationally and with respect to the information in our data set infeasible.

5. I could not open the zenodo link (10.5281/zenodo.8264102) provided to scripts used to perform the analysis. Additionally, the authors mention that data analysed will only be provided upon request. In the spirit of reproducibility, I encourage the authors to publish both the scripts and data. An episet_id for GISAID or Genbank accession would be sufficient.

We apologize for the inconvenience with the Zenodo link. We double checked the code availability and were able to access it via DOI (full URL: <https://zenodo.org/records/8264102>). The newest version is now available under <https://zenodo.org/records/8270192>, leading also to a changed DOI in the manuscript. Moreover, we now published the variant sequences under <https://www.ncbi.nlm.nih.gov/sra/PRJNA1017685> and changed the data sharing section of the manuscript accordingly.

Changes in the manuscript:

Initial submission:

Data sharing

Data will be made available to other researchers upon qualified request.

Revised submission:

Data availability

The models and population average data are available at Zenodo³. The variant sequences are published in the Sequence Read Archive⁴ under project number PRJNA1017685. Individual level data will be made available to other researchers upon qualified request.

Code availability

The code for model creation and data aggregation is available at Zenodo³.

Response to Reviewer #2

Merkt et al. analyse antibody levels against SARS-CoV-2 and genetic sequencing data from two locations in Ethiopia. These are useful data, particularly as they span much of the pandemic period, and overall the analysis has been well executed.

We thank the reviewer for this positive evaluation.

The antibody data come from serosurveys conducted in two locations (Addis Ababa and Jimma) and in two different populations (healthcare workers and community members). In their primary analysis, the authors focus on community members and combine data from the 2 locations. Focusing on community members seems reasonable, however, it would be useful to also present data separately for each location. It is probable that seroprevalence trends vary between the sites, and an exploration of this variation would be informative.

We thank the reviewer for this suggestion. The revised version of the manuscript includes a site-specific analysis. The information is included in the supplementary notes as new Supplementary Notes Figure. Overall, we find similar trends for all sites, with small time shifts.

Changes in the supplementary notes:

Initial submission:

-

Revised submission:

Figure SN4new. Antibody data of community members and healthcare workers by site of collection.

In addition to the statistical analysis, the study uses the data (antibody and sequencing data) to fit two models of disease transmission. While these models appear reasonable, they do not align well with the PCR positivity data. They fail to capture two prominent peaks in infection and predict a significant peak where none exists. This discrepancy suggests that these model results may have limited predictive value. Emphasizing the weaknesses of these models is essential.

We thank the reviewer for pointing out that the datasets (and their limitations) as well as the model fit needs additional discussion.

We think that the discrepancies identified by the reviewer indicate rather a limitation of the datasets than the model. We obtained antibody and virus strain data for Addis Ababa and Jimma, but the PCR test positivity rates for Ethiopia as a whole. This causes smaller inconsistencies, amongst other things, the PCR test positivity rates have two peaks which are not captured by the antibody data. One would expect that antibody data rise briefly after the peaks, but this does not happen. If we e.g. compare antibody data and national test positivity rate closely at the first peak (February 2021), we can see that our antibody data already plateaus while the national test positivity rate transitions to a second peak (May 2021). This could be explained by a major outbreak (or a delayed wave) in a different part of Ethiopia which is not covered in our antibody data (c.f. Figure RL2). Moreover the availability of tests, especially in the countryside is rather low. Thus the selection of persons tested and the availability of tests can influence the rates and there might be an additional bias that we cannot account for. Hence, conclusions from data such as serology seem more reliable as this is clearly substantiated evidence.

In early versions of the models we did not include the national PCR test positivity rates. Yet, we observed that these data are helpful to provide some information about the overall number of cases, in later waves (e.g the omicron waves) where antibody prevalence already reached up to 100%.

Moreover, the future predictions on accumulated antibody levels of the antibody prevalence model, are only to a certain extend influenced by the exact timing of peaks. However, for the retrospective analysis of infection peaks for different vaccination dosages, it causes considerable uncertainty.

Figure RL2: Zoomed in comparison of antibody levels and national test positivity rates.

We highlight these limitations now in our Discussion section and extended the respective presentations in the result section of the model. The latter changes are presented at the corresponding comments below.

Changes in the manuscript:

Initial submission:

On the modeling side, the combination of antibody and variant data from Addis Ababa and Jimma with national test positivity rates might be criticized. The latter was done to provide information about the overall number of cases. Despite its limitations, this study provides an unprecedented insight into the dynamics of COVID-19 infections over time and the impact of the variants in Ethiopia.

Revised submission:

The models we propose here are based on antibody and variant data from Addis Ababa and Jimma, as well as nation-wide test-positivity rates. While the sampling regions in Addis Ababa and Jimma cover areas of different population density and should prove a comprehensive picture, they might not be fully representative for the spread of SARS-CoV-2 in Ethiopia. An indication for this is that the nation-wide test-positivity rate increases in April, 2021 and January, 2022, while the antibody data do not show substantial changes at these time points or briefly afterwards. Hence, the use of the combined dataset for the assessment of Ethiopia is an extrapolation. Moreover, (i) the description of cross-immunity factors as a function solely depending on MOIC neglects that other mutations might also affect immune escape potential, (ii) the dependency of cross-immunity after infections with different variants on the union of mutations from previous variants might overemphasize later variants (since secondary infections are assumed to mainly recall cross-reactive antibodies). Yet, these simplifications were important to ensure computational feasibility and balance model complexity and statistical power in the data. A consideration of all mutations would have increased the number of model parameters by a factor of 9.5 and the dataset would have been insufficient to inform them. Despite its limitations, this study provides an unprecedented insight into the dynamics of COVID-19 infections over time and the impact of the variants in Ethiopia.

l.68 “Given our data, serological evidence of past infection in Ethiopia alone suggests more than ten times as many infections by autumn 2022.” Ten times as many as the number reported?

We apologize for the imprecise statement. It is correct that we were referring to ten times as many as past infection as the number reported. We clarified this by implementing the following changes to the manuscript.

Changes in the manuscript:

Initial submission:

Given our data, serological evidence of past infection in Ethiopia alone suggests more than ten times as many infections by autumn 2022².

Revised submission:

Given our data, serological evidence of past infection suggests that by autumn 2022 there were ten times as many infections in Ethiopia as officially reported².

l. 80 “leading to adjustments in vaccination policies in various hospitals.” More detail needed.

We agree with the reviewer that more detail would be needed to make this statement meaningful. Since the investigation of vaccination policies is subject of so far unpublished work we decided to rather remove it from the current manuscript.

Changes in the manuscript:

Initial submission:

By employing epidemiological modeling, we predicted prevalence levels above 50% for the population, leading to adjustments in vaccination policies in various hospitals.

Revised submission:

By employing epidemiological modeling, we predicted prevalence levels above 50% for the population.

l. 85 “Previous publications touch upon this topic hypothetically”. More detail needed.

We added more detail through the following changes:

Changes in the manuscript:

Initial submission:

Previous publications touch upon this topic hypothetically and only very recently longitudinal data from Ethiopia has become available⁵.

Revised submission:

Previous publications touch upon this topic hypothetically, e.g. Gudina et al. by simulating a scenario with two variants², but longitudinal data on variant distribution has only recently become available for Ethiopia⁵.

l.109 “We utilized a second epidemiological model to predict future antibody dynamics, providing insights into the expected long-term immunity landscape in the Ethiopian population, which is critical to prevent massive spread of SARS-CoV-2 and potential health system crisis in the future.” It is a strong claim that the model will prevent spread of SARS-CoV-2 and avert a health crisis! Consider rephrasing or deleting this sentence.

We apologize for this unfortunate choice of words. We update the statement:

Changes in the manuscript:

Initial submission:

We utilized a second epidemiological model to predict future antibody dynamics, providing insights into the expected long-term immunity landscape in the Ethiopian population, which is critical to prevent massive spread of SARS-CoV-2 and potential health system crisis in the future.

Revised submission:

We utilized a second epidemiological model to predict future antibody dynamics, providing insights into the expected long-term immunity landscape in the Ethiopian population. This might provide decision makers with information which is helpful for the assessment of the situation and the choice of appropriate measures.

Table 1. The high % women (75%) in Addis Ababa in R1 and R2 should be explained. Also, it would be helpful to add the dates of each survey round to the table (could be as a footnote).

The percentage of women in Addis Ababa for Round 1 and 2 indeed is rather high. The reason is that for organisational reasons we were in the antibody study limited to convenience sampling. Yet, for our previous study we analysed the difference of antibody prevalence with respect to sex and could not find any significant difference. Moreover, through sampling participants in densely and sparsely populated areas of Addis Ababa (Addis Ketema resp. Yeka) and restricting participants to one person per household to ensure a broad coverage.

In response to this comment as well as a comment on line 621 below (where we highlight this change) we included more information about the study design into this manuscript (in addition to referring to our previous publication). Moreover, we now discuss the issue of representativeness in the Discussion section and added survey dates to Table 1.

Changes in the manuscript:

Initial submission:

On the modeling side, the combination of antibody and variant data from Addis Ababa and Jimma with national test positivity rates might be criticized. The latter was done to provide information about the overall number of cases. Despite its limitations, this study provides an unprecedented insight into the dynamics of COVID-19 infections over time and the impact of the variants in Ethiopia.

Revised submission:

The models we propose here are based on antibody and variant data from Addis Ababa and Jimma, as well as nation-wide test-positivity rates. While the sampling regions in Addis Ababa and Jimma cover areas of different population density and should prove a comprehensive picture, they might not be fully representative for the spread of SARS-CoV-2 in Ethiopia. An indication for this is that the nation-wide test-positivity rate increases in April, 2021 and January, 2022, while the antibody data do not show substantial changes at these time points or briefly afterwards. Hence, the use of the combined dataset for the assessment of Ethiopia is an extrapolation. Moreover, (i) the description of cross-immunity factors as a function solely depending on MOIC neglects that other mutations might also affect immune escape potential, (ii) the dependency of cross-immunity after infections with different variants on the union of mutations from previous variants might overemphasize later variants (since secondary infections are assumed to mainly recall cross-reactive antibodies). Yet, these simplifications were important to ensure computational feasibility and balance model complexity and statistical power in the data. A consideration of all mutations would have increased the number of model parameters by a factor of 9.5 and the dataset would have been insufficient to inform them. Despite its limitations, this study provides an unprecedented insight into the dynamics of COVID-19 infections over time and the impact of the variants in Ethiopia.

Changes in the manuscript (continued):

Initial submission:

Table 1: Demographic characteristics of community members participating in study. Age denoted as median and 90% quantiles, and sex in absolute and relative numbers.. Round 1-3 (R1-R3) are the previous study².

	Jimma					Addis Ababa				
	R1	R2	R3	R4	R5	R1	R2	R3	R4	R5
Participants	536	325	267	539	575	361	314	721	424	461
Age	30 (19, 63)	30 (19, 62)	32 (19, 63)	33 (20, 65)	32 (19, 63)	36 (21, 68)	36 (22, 67)	35 (21, 67)	33 (19, 65)	38 (20, 68)
Sex										
Women	260 (48.5%)	166 (51.1%)	136 (50.9%)	331 (61.4%)	69 (12.0%)	279 (77.3%)	236 (75.2%)	360 (49.9%)	209 (49.3%)	162 (35.1%)
Men	276 (51.5%)	159 (48.9%)	131 (49.1%)	207 (38.4%)	65 (11.3%)	79 (21.9%)	70 (22.3%)	109 (15.1%)	71 (16.7%)	299 (64.9%)
Missing	0 (0.0%)	0 (0.0%)	0 (0.0%)	1 (0.2%)	441 (76.7%)	3 (0.8%)	8 (2.5%)	252 (35.0%)	144 (34.0%)	0 (0.0%)
Anti-N positive	139 (25.9%)	114 (35.1%)	107 (40.1%)	313 (58.1%)	543 (94.4%)	165 (45.7%)	150 (47.8%)	234 (32.5%)	286 (67.5%)	458 (99.3%)
Vaccinated	0 (0.0%)	0 (0.0%)	1 (0.4%)	47 (8.7%)	195 (33.9%)	0 (0.0%)	0 (0.0%)	0 (0.0%)	28 (6.6%)	167 (36.2%)

Revised submission:

Table 1: Demographic characteristics of community members participating in study. Age denoted as median and 90% quantiles, and sex in absolute and relative numbers.. Round 1-3 (R1-R3) are the previous study².

	Jimma					Addis Ababa				
	R1 (Dec 20)	R2 (Jan 21)	R3 (Feb 21)	R4 (Aug 21)	R5 (Apr 22)	R1 (Jan 21)	R2 (Feb 21)	R3 (Apr 21)	R4 (Sep 21)	R5 (Mar 22)
Participants	536	325	267	539	575	361	314	721	424	461
Age	30 (19, 63)	30 (19, 62)	32 (19, 63)	33 (20, 65)	32 (19, 63)	36 (21, 68)	36 (22, 67)	35 (21, 67)	33 (19, 65)	38 (20, 68)
Sex										
Women	260 (48.5%)	166 (51.1%)	136 (50.9%)	331 (61.4%)	69 (12.0%)	279 (77.3%)	236 (75.2%)	360 (49.9%)	209 (49.3%)	162 (35.1%)
Men	276 (51.5%)	159 (48.9%)	131 (49.1%)	207 (38.4%)	65 (11.3%)	79 (21.9%)	70 (22.3%)	109 (15.1%)	71 (16.7%)	299 (64.9%)
Missing	0 (0.0%)	0 (0.0%)	0 (0.0%)	1 (0.2%)	441 (76.7%)	3 (0.8%)	8 (2.5%)	252 (35.0%)	144 (34.0%)	0 (0.0%)
Anti-N positive	139 (25.9%)	114 (35.1%)	107 (40.1%)	313 (58.1%)	543 (94.4%)	165 (45.7%)	150 (47.8%)	234 (32.5%)	286 (67.5%)	458 (99.3%)
Vaccinated	0 (0.0%)	0 (0.0%)	1 (0.4%)	47 (8.7%)	195 (33.9%)	0 (0.0%)	0 (0.0%)	0 (0.0%)	28 (6.6%)	167 (36.2%)

Changes in the supplement:

Initial submission:

Supplementary Table 1: Demographic characteristics of healthcare workers study participants. Age denoted as median and 90% quantiles. Round 1-3 (R1-R3) are the previous study of Gudina et al 2021.

	Jimma Medical Center					St Paul's Hospital				
	R1	R2	R3	R4	R5	R1	R2	R3	R4	R5
Participants	510	434	372	508	510	461	284	116	176	196
Age	26 (22, 39)	26 (23, 41)	26 (23, 39)	28 (21, 39)	29 (23, 50)	28 (22, 42)	28 (20, 42)	26 (20, 42)	26 (21, 42)	30 (23, 40)
Sex										
Women	271 (53.1%)	231 (53.2%)	199 (53.5%)	273 (53.7%)	68 (13.3%)	236 (51.2%)	103 (36.3%)	44 (37.9%)	92 (52.3%)	4 (2.0%)
Men	239 (46.9%)	203 (46.8%)	173 (46.5%)	233 (45.9%)	45 (8.8%)	222 (48.2%)	76 (26.8%)	30 (25.9%)	56 (31.8%)	4 (2.0%)
Missing	0 (0.0%)	0 (0.0%)	0 (0.0%)	2 (0.4%)	397 (77.8%)	3 (0.7%)	105 (37.0%)	42 (36.2%)	28 (15.9%)	188 (95.9%)
Anti-N positive	157 (30.8%)	198 (45.6%)	209 (56.2%)	364 (71.7%)	490 (96.1%)	40 (8.7%)	112 (39.4%)	60 (51.7%)	128 (72.7%)	189 (96.4%)
Vaccinated	0 (0.0%)	0 (0.0%)	0 (0.0%)	217 (42.7%)	149 (29.2%)	1 (0.2%)	1 (0.4%)	0 (0.0%)	71 (40.3%)	5 (2.6%)

Revised submission:

Supplementary Table 1. Demographic characteristics of healthcare workers study participants. Age denoted as median and 90% quantiles. Round 1-3 (R1-R3) are the previous study of Gudina et al 2021.

	Jimma Medical Center					St Paul's Hospital				
	R1 (Nov 20)	R2 (Dec 20)	R3 (Feb 21)	R4 (Aug 21)	R5 (Apr 22)	R1 (Aug 20)	R2 (Dec 20)	R3 (Feb 21)	R4 (Sep 21)	R5 (Apr 22)
Participants	510	434	372	508	510	461	284	116	176	196
Age	26 (22, 39)	26 (23, 41)	26 (23, 39)	28 (21, 39)	29 (23, 50)	28 (22, 42)	28 (20, 42)	26 (20, 42)	26 (21, 42)	30 (23, 40)
Sex										
Women	271 (53.1%)	231 (53.2%)	199 (53.5%)	273 (53.7%)	68 (13.3%)	236 (51.2%)	103 (36.3%)	44 (37.9%)	92 (52.3%)	4 (2.0%)
Men	239 (46.9%)	203 (46.8%)	173 (46.5%)	233 (45.9%)	45 (8.8%)	222 (48.2%)	76 (26.8%)	30 (25.9%)	56 (31.8%)	4 (2.0%)
Missing	0 (0.0%)	0 (0.0%)	0 (0.0%)	2 (0.4%)	397 (77.8%)	3 (0.7%)	105 (37.0%)	42 (36.2%)	28 (15.9%)	188 (95.9%)
Anti-N positive	157 (30.8%)	198 (45.6%)	209 (56.2%)	364 (71.7%)	490 (96.1%)	40 (8.7%)	112 (39.4%)	60 (51.7%)	128 (72.7%)	189 (96.4%)
Vaccinated	0 (0.0%)	0 (0.0%)	0 (0.0%)	217 (42.7%)	149 (29.2%)	0 (0.0%)	0 (0.0%)	0 (0.0%)	71 (40.3%)	5 (2.6%)

I.136 “Our analysis revealed that in the majority of individuals, Anti-S and Anti-N antibodies were present (Figure 1A-E), suggesting an infection event.” Do you mean the % with non-zero values for these 2 assays. What are these %?

Indeed we meant the % with non-zero values, in the sense of above the positivity thresholds of the antibody tests, for both Anti-S and Anti-N. We now clarified the wording and included the % numbers also in the text.

Changes in the manuscript:

Initial submission:

Our analysis revealed that in the majority of individuals, Anti-S and Anti-N antibodies were present (Figure 1A-E), suggesting an infection event.

Revised submission:

Our analysis revealed that in April 2022 the majority of individuals (in Round 5: 95.9% of the healthcare workers and 94.8% of the community members), tested positive for Anti-S and Anti-N antibodies (Figure 1A-E), suggesting an infection event.

Figure 1. This figure is very informative. If I understand correctly, the cut-off values were chosen by applying 1-dimensional k-means separately to anti-N and anti-S antibody levels. Did you try clustering using 2-dimensional k means, i.e. applying k means to anti-N and anti-S simultaneously?

The cut-off values were indeed calculated by applying 1-dimensional k-means separately. We did also try k-means clustering to the two dimensional set of all antibodies which were above the reactivity threshold for each Anti-S and Anti-N. Here, we encountered the problem that the two modes for Anti-S which are clearly depicted in the histograms are not precisely represented by the boundaries between clusters (c.f. Response Letter Figure RL3). Moreover, we could not find an appropriate and straightforward way to convert those boundaries into cutoff values for aggregated Anti-S and Anti-N, which we need for example for the multivariate model where we only included Anti-S information. Finally, the one dimensional data sets have a slightly higher statistical power, since for some study participants only one of the antibody tests, Anti-N or Anti-S, was successful.

Figure RL3: Result of 4-means clustering of 2-dimensional reactive antibody data

In order to emphasize the reasoning for our choice we added the following to the manuscript:

Changes in the manuscript:

Initial submission:

Employing k-means clustering with two means on the S-positive samples from all five rounds, we determined the cutoff value for the groups with one or multiple exposures to be 274.5 (for more details see Supplementary Information's Supplementary Note 1 and Figure SN2).

Revised submission:

Employing **1-dimensional** k-means clustering with two means on the S-positive samples from all five rounds, we determined the cutoff value for the groups with one or multiple exposures to be 274.5 (for more details see Supplementary Information's Supplementary Note 1 and Figure SN2).

Changes in the supplement:

Initial submission:

We utilized scikit-learn's k-means clustering implementation to categorize the remaining data points above the threshold into two distinct groups⁶. The midpoint between the two resulting groups' centers was determined as the separation value.

Revised submission:

We utilized scikit-learn's k-means clustering implementation to categorize the remaining data points above the threshold into two distinct groups⁶. **We chose clustering the antibody datasets separately, i.e. 1-dimensional clustering, motivated by the bi-modal distributions we observed in the histograms for Anti-S. Moreover, the separate clustering of the Anti-N or Anti-S data provides: (i) a slightly higher statistical power, since for some study participants only one the antibody tests, Anti-N or Anti-S, was successful; and (ii) clear cutoff values for aggregated Anti-S measurements (e.g. by using midpoint of the two resulting groups' centers), which is necessary for the multivariate model.**

l.144 "Interestingly, most individuals vaccinated also showed reactivity for Anti-N" Please provide the percentage to support this statement.

We include the percentage in the revised version of the manuscript.

Changes in the manuscript:

Initial submission:

Interestingly, most individuals vaccinated also showed reactivity for Anti-N, suggesting they had been exposed to the infection prior to or shortly after vaccination.

Revised submission:

Interestingly, most individuals vaccinated also showed reactivity for Anti-N (**in Round 5: 94.8% of the healthcare workers and 96.4% of the community members**), suggesting they had been exposed to the infection prior to or shortly after vaccination.

Figure 3. The model fits the antibody data but it does not fit data on the national PCR positivity rate - it misses two peaks and predicts a significant peak when there is none. This suggests that either the model predictions or the accuracy of the PCR data, or possibly both, should be viewed with caution.

Here we refer to our general discussion of antibody data versus national PCR positivity rates above and implemented the following addition to the results section:

Changes in the manuscript:

Initial submission:

The antibody levels and variant distributions (the primary focus of our investigation) are captured accurately. The national test positivity rate is described well up to two peaks (which might be caused by different regions in Ethiopia).

Revised submission:

The antibody levels and variant distributions (the primary focus of our investigation) are captured accurately. The national test positivity rate is described well up to two peaks (which might be caused by different regions in Ethiopia). **In fact looking at the timing of the first peak, which is missed by our model, we see that our antibody data is already saturated and hence tells a different story than the nationally reported data.**

Figure 5. Again the model does predict the PCR positivity rate very well.

Here we refer to our general discussion of antibody data versus national PCR positivity rates above and implemented the following addition to the results section:

Changes in the manuscript:

Initial submission:

As the antibody-level model provides an accurate description of the available data, we used it to predict the current antibody levels, including observations of antibody levels until April 2022.

Revised submission:

As for the multivariant model there is some discrepancy between national test positivity rate and model description (which might be caused by different regions in Ethiopia). Nevertheless, the antibody-level model provides an accurate description of the available antibody data, so that we used it to predict the current antibody levels, including observations of antibody levels until April 2022.

1.621 It would be helpful to provide a brief description of how participants were recruited into the survey rather than just refer to Gudina et al. 2021. The description should include details such as whether it was a convenience sample, the age range of participants, any exclusion criteria, and how data on vaccination status were acquired.

We thank the reviewer for this suggestion and included more details on recruitment of study participants directly in to the manuscript.

Changes in the manuscript:

Initial submission:

Community members and healthcare workers were recruited for the serology study (see in-depth description in Gudina et al. 2021²). In Addis Ababa, community members from Addis Ketema and Yeka sub cities were recruited. In Jimma, no specific region was chosen and rural participants were recruited around the Jimma Zone. One participant per household was sampled to avoid any clustering effects.

Revised submission:

Community members and healthcare workers were recruited for the serology study based on convenience sampling. Hospital workers including clinical staff, medical interns, cleaners, guards, food handlers, and administrative personnel—were recruited at two hospitals, the St Paul’s Hospital in Addis Ababa and the Jimma Medical Center. In Addis Ababa, community members from Addis Ketema and Yeka sub cities were recruited. In Jimma, no specific region was chosen and rural participants were recruited around the Jimma Zone. Sample size were initially calculated in July, 2020, when not much baseline data was available and later became flexible as more data became available. Moreover, as the rate of dropout was more than 30% (our initial expectation), we recruited more participants to compensate for the dropouts (c.f. Supplementary Figure 1 for detailed studyflow). One participant per household was sampled to avoid any clustering effects and households were selected randomly in a way that avoided frequent interaction from the next candidate household to prevent cross-contamination. Overall the median age was 30 with 90th percentile (20,60) and 55.6% of participants which provided information about sex were female (for round and site specific demographics see Table 1 and Supplementary Table 1). All participants of the first 3 rounds were enrolled before the introduction of COVID-19 vaccines in Ethiopia. In later rounds participants provided their vaccination status and dates through a questionnaire. For more details see in-depth description in Gudina et al. 2021².

Supplementary Note 2: The description of the model in Supplementary Note 2 could be made clearer with more details. E.g. Does the variant index $i = 1, \dots, 8$ correspond to the columns of Table SN2, i.e. 1=wildtype, 2=wildtype* etc.? Why are the 3rd and 4th infections limited to $k=7, 8$ (omicron?)

We agree that more details here could be beneficial and added the following to Supplementary Note 2:

Changes in the supplement:

Initial submission:

We utilize the SEIR (susceptible, exposed, infectious, and recovered) framework as basis for our model structure. For $i = 1, \dots, 8$ representing the variant index we have the following equations for first infection or vaccination

...

where P_i is the set of potential reinfections after infection with variant i , described by Table SN2 where vaccination is treated as previous infection with the wildtype variant. Furthermore \hat{I}_i is the sum of all currently infected with variant i , N the sum of all state variables, t_{0i} the entrance date of variant i and v_k denote the k -th vaccination rates.

Revised submission:

We utilize the SEIR (susceptible, exposed, infectious, and recovered) framework as basis for our model structure. Assuming a maximum number of 4 infections all combinations of our 8 variants would lead to a system of $8^4 = 4096$ pathways. Hence in order to obtain a computationally feasible system while still retaining realism we exclude pathways which deviated from the chronological order of variant appearances worldwide. We define by P_i the set of potential reinfections after infection with variant i , described by Table SN2 where vaccination is treated as previous infection with the wildtype variant. Furthermore to account for the reported inter-infection intervals we assume third infections before omicron played a negligible role and allow a fourth infection only for omicron BA.4/5, i.e. P_i collapses to $\{7, 8\}$ resp. $\{8\}$. For $i = 1, \dots, 8$ representing the variant index, where these numbers correspond to columns in Table SN2, we have the following equations for first infection or vaccination

...

where \hat{I}_i is the sum of all currently infected with variant i , N the sum of all state variables, t_{0i} the entrance date of variant i and v_k denote the k -th vaccination rates.

1.465 “This suggests that a substantial percentage... had already experienced a second or even third infection” Please provide the actual percent of 2nd and 3rd infections?

Changes in the manuscript:

Initial submission:

This suggests that a substantial percentage of the 120 million inhabitants of Ethiopia had already experienced a second or even third infection by the time of the last sampling round in April 2022.

Revised submission:

This suggests that by the end of the last sampling round in April 2022, already 55.1% of the inhabitants of Ethiopia recovered from two SARS-CoV-2 infections. Another 4.1% of the inhabitants of Ethiopia recovered from three SARS-CoV-2 infections.

1.505 “For pooled protection against ancestral variants, they obtained protection levels of 84.9% (72.8, 91.8). Comparing their result (95% CI) to our findings (with 90% CIs) of 90.0% (85.3, 94.9) of wildtype and wildtype* against themselves and 75.5% (72.2, 78.7) against each other,” Why are you restricting to wildtype and not provide an estimate of protection against ancestral variants? It appears that you are not comparing like with like.

The COVID-19 Forecasting Team⁷ used “ancestral variants” as a collective term referring to

all variants which occurred earlier than the alpha variant. In our context this corresponds to wildtype and wildtype*. Accordingly, we think that we are comparing here the appropriate quantities. We apologize for the unclear wording and tried to clarify it in the following way:

Changes in the manuscript:

Initial submission:

For pooled protection against ancestral variants, they obtained protection levels of 84.9% (72.8, 91.8). Comparing their result (95% CI) to our findings (with 90% CIs) of 90.0% (85.3, 94.9) of wildtype and wildtype* against themselves and 75.5% (72.2, 78.7) against each other, we see that our estimates lay well inside the study's CI (Figure 4E, Supplementary Figure 2).

Revised submission:

For pooled protection against ancestral variants, which the COVID-19 Forecasting Team uses as a collective term for all variants which occurred earlier than the alpha variant, they obtained protection levels of 84.9% (72.8, 91.8). Comparing their result (95% CI) to our findings (with 90% CIs) of 90.0% (85.3, 94.9) of wildtype and wildtype*, which in our context corresponds to variants earlier than alpha variant, against themselves and 75.5% (72.2, 78.7) against each other, we see that our estimates lay well inside the study's CI (Figure 4E, Supplementary Figure 2).

L.508 “Overall protection against the alpha variant” Please clarify what is mean by “overall protection” - is it the average protection against infection with the alpha variant among those who have previously been infected with any variant?

“Overall protection” is used here interchangeably with the “pooled protection” against all previous variants obtained by the cited COVID-19 Forecasting Team in their Bayesian meta analysis of 65 studies⁷. To clarify this also in the manuscript we changed the wording here and stick now to the “pooled protection”. Moreover we added more information about the analysis method of the Covid Forecasting Team.

Changes in the manuscript:

Initial submission:

The estimates and predictions provide an in-depth assessment of the situation in Ethiopia. On the high level, they also agree with other studies, including the meta-analysis by the COVID-19 Forecasting Team, which pooled analysis results of 65 studies from 19 different countries⁷.

...

Overall protection against the alpha variant is stated to be 90.0% (54.8, 98.4) while our values range from 53.1% (50.2, 56.0) to 90.0% (85.3, 94.9) (Figure 4E, column on alpha variant), where our lowest median is only slightly below their CI's lower bound and the CIs overlap (Supplementary Figure 2).

Revised submission:

The estimates and predictions provide an in-depth assessment of the situation in Ethiopia. On the high level, they also agree with other studies, including the meta-analysis by the COVID-19 Forecasting Team, which used Bayesian meta-regression to pool results of 65 studies from 19 different countries on protection against new variants by past infections with earlier variants⁷.

...

Pooled protection against the alpha variant is stated to be 90.0% (54.8, 98.4) while our values range from 53.1% (50.2, 56.0) to 90.0% (85.3, 94.9) (Figure 4E, column on alpha variant), where our lowest median is only slightly below their CI's lower bound and the CIs overlap (Supplementary Figure 2).

Response to Reviewer #3

The authors need to specify the type of vaccines being administered in Ethiopia, since certain vaccine platforms may have stronger immunogenicity (mRNA) than others (inactivated/vector based), all of which were used in the region of Africa. It's possible that the stronger immunogenicity of mRNA vaccine with respect to natural infection could also potentially create bi-modal pattern of the distribution of the anti-S antibody level, in addition to repeated infections. It would be great if the authors could show the range of antibody level for individuals without prior infection but vaccination only, using the same assay of the study, and how it compare with the anti-S antibody level in this cohort population.

We did not collect data for type of the vaccines during the Round 4 (August-September 2021) as all vaccines in Ethiopia by then were only Covishield (AstraZeneca type vaccine manufactured by Serum Institute of India). By Round 5 (March-April 2022) Johnson & Johnson (J & J) has become another major type of the vaccine. Although few doses of Sinovac/Sinopharm, Sputnik-V, Moderna, and Pfizer-BioNTech were reported to be donated to the country, they were very little and hence negligible. In the current (unpublished) COVICIS data, we have been collecting detailed vaccination information (Response Letter Table RL1).

Summing up the vaccine data based on this information, we see that the vast majority of vaccines administered in Ethiopia among healthcare workers and community members were either Covishield or J&J, while the use of mRNA vaccines is negligible.

We included this information and reasoning into the Results section.

Vaccine Type	First Dose	Second Dose	Third Dose	Overall (total)
Covishield (AstraZeneca)	81	193	54	328 (66.8%)
Johnson & Johnson	112	15	16	143 (29.1%)
Other	0	9	11	20 (4.1%)

Table RL1: Distribution of vaccine types from COVICIS study.

Moreover, the information about individuals without infection but vaccination only is contained in Figure 1A-E and Supplementary Figure 3A-E, where Anti-S levels are plotted against Anti-N levels for community members and healthcare workers respectively. Looking at Anti-S levels for negative Anti-N response, hence no previous infection, one can see that they cover the same range as non-vaccinated but infected individuals. In particular in Round 4 for healthcare workers, where most are already vaccinated but still a substantial number Anti-N negative. Figure RL4 highlights this in a comprehensive way.

Comparison of Anti-S levels from vaccine vs infection (Community members)

Comparison of Anti-S levels from vaccine vs infection (Healthcare workers)

Figure RL4: Distributions of Anti-S levels for vaccinated with Anti-N negative and unvaccinated with Anti-N positive.

Changes in the manuscript:

Initial submission:

As large-scale vaccination campaigns started in Ethiopia rather late in November 2021, the data suggests that sampling in round three coincided with waves of SARS-CoV-2 infections. First confirmed vaccinated individuals show up only in rounds four (August 2021, Figure 1D) and five (April 2022, Figure 1E). Interestingly, most individuals vaccinated also showed reactivity for Anti-N, suggesting they had been exposed to the infection prior to or shortly after vaccination.

Revised submission:

As large-scale vaccination campaigns started in Ethiopia rather late in November 2021, the data suggests that sampling in round three coincided with waves of SARS-CoV-2 infections. First confirmed vaccinated individuals show up only in rounds four (August 2021, Figure 1D) and five (April 2022, Figure 1E). Interestingly, most individuals vaccinated also showed reactivity for Anti-N (in Round 5: 94.8% of the healthcare workers and 96.4% of the community members), suggesting they had been exposed to the infection prior to or shortly after vaccination. By Round 4 all vaccines in Ethiopia were Covishield (AstraZeneca type vaccine manufactured by Serum Institute of India) and by Round 5 Johnson & Johnson has become another major type of the vaccine. Although few doses of Sinovac/Sinopharm, Sputnik-V, Moderna, and Pfizer-BioNTech were reported to be donated to the country, they were very little and hence negligible. Therefore we can safely disregard the influence of mRNA vaccines in our study.

Changes in the supplement:

Initial submission:

-

Revised submission:

There was no vaccine publicly available in Ethiopia until after Round 3¹. Because of this information about general vaccine availability in combination with our previous observation that vaccinated individuals are more likely to answer questions on the vaccination status on the questionnaire than unvaccinated individuals, we considered individuals without an answer (“N/A”) as “unvaccinated” for modelling. This is also supported by official nation wide numbers of people with at least one dose of vaccine, provided by Our World in Data (ourworldindata.org) and depicted in Figure SN3new. Moreover, we treat the effect of vaccine and infection on Anti-S levels analogously. This is based on the comparison of the observed antibody levels for healthcare workers (Supplementary Figure 1) and community members (Figure 1). There from Round 3 to Round 4 for healthcare workers a clear shift from medium Anti-S to high Anti-S is observed in response to vaccination, but community members reach the same levels by infections alone.

Figure SN3new. Histograms of distributions of vaccination information from study participants at each round. “N/A” responses before public availability of vaccine in Ethiopia are highlighted by hatching. For community members official, national vaccination numbers (provided by Our World in Data) are indicated in red above each round and percentages from our data set of “N/A” responses after public availability of vaccines in Ethiopia are displayed inside of the corresponding bars.

Line 281-283: The authors use high-level anti-S antibody to infer individuals with at least two infections, based on the bi-model distribution of the anti-S antibody titer distribution as shown in Figure SN1-SN2. However, if we look at Figure SN1, it seems like having vaccination significantly increase the chances of being in the high titer level (the more doses of vaccination, the stronger the bias towards high titer). So it’s likely that individuals with only one prior infection with prior vaccination(s) could also end up in the high anti-S antibody level. How would the authors be able to differential repeated infections with hybrid immunity (prior infection and vaccination) purely based on S titer level?

We agree with the reviewer that it is not possible to distinguish multiple infections from a mixture of infections and vaccines from Anti-S data only. Indeed – although it did not become clear – we did not assume this. The multivariant model implements an observable mapping which maps multiple infections, multiple vaccinations and mixtures of both into one observable

(c.f. “Antibody levels high” in Figure 3B and observable A_1 in Supplementary Note 2). For the antibody level this does not play a role since here Anti-S and Anti-N data are implemented in combination. We apologize for the unclear wording and made this more precise by including to following additions to our manuscript:

Changes in the manuscript:

Initial submission:

To assess the evolution of the SARS-CoV-2 pandemic in Ethiopia, we parameterized the multivariant model using data on antibody levels, viral variant distribution, and national test positivity rate. The Anti-S antibody measurements were used to provide information on the fraction of individuals with a single infection (medium level) and the fraction of individuals with at least two infections (high level).

Revised submission:

To assess the evolution of the SARS-CoV-2 pandemic in Ethiopia, we parameterized the multivariant model using data on antibody levels, viral variant distribution, and national test positivity rate. The Anti-S antibody measurements were used to provide information on the fraction of individuals with a single infection **or vaccination** (medium level) and the fraction of individuals with at least two infections **, vaccinations or a combination of both** (high level). **Since it is impossible to distinguish between vaccinations and infections from Anti-S levels we implemented observables corresponding to the medium and high levels without discriminating between vaccination or infection (c.f. Supplementary Note 2 for detailed equations).**

Line 262: Here the authors consider the protective immunity profile for individuals with more than one prior infections as a simple union of the mutations from the previous variants. However, it’s evident now that the ordering of exposures may bias the immune response due to immune memory (imprinting). I.e, secondary infection largely recall cross-reactive antibodies induced by previous variant, with very limited response towards novel epitopes of variant that caused the secondary infection. This shall be discussed as a limitation of the model.

We thank the reviewer for this suggestion and included the following addition to the Discussion section:

Changes in the manuscript:

Initial submission:

On the modeling side, the combination of antibody and variant data from Addis Ababa and Jimma with national test positivity rates might be criticized. The latter was done to provide information about the overall number of cases. Despite its limitations, this study provides an unprecedented insight into the dynamics of COVID-19 infections over time and the impact of the variants in Ethiopia.

Revised submission:

The models we propose here are based on antibody and variant data from Addis Ababa and Jimma, as well as nation-wide test-positivity rates. While the sampling regions in Addis Ababa and Jimma cover areas of different population density and should prove a comprehensive picture, they might not be fully representative for the spread of SARS-CoV-2 in Ethiopia. An indication for this is that the nation-wide test-positivity rate increases in April, 2021 and January, 2022, while the antibody data do not show substantial changes at these time points or briefly afterwards. Hence, the use of the combined dataset for the assessment of Ethiopia is an extrapolation. Moreover, (i) the description of cross-immunity factors as a function solely depending on MOIC neglects that other mutations might also affect immune escape potential, (ii) the dependency of cross-immunity after infections with different variants on the union of mutations from previous variants might overemphasize later variants (since secondary infections are assumed to mainly recall cross-reactive antibodies). Yet, these simplifications were important to ensure computational feasibility and balance model complexity and statistical power in the data. A consideration of all mutations would have increased the number of model parameters by a factor of 9.5 and the dataset would have been insufficient to inform them. Despite its limitations, this study provides an unprecedented insight into the dynamics of COVID-19 infections over time and the impact of the variants in Ethiopia.

Response Letter References

1. WHO. *Ethiopia launches a COVID-19 vaccination campaign targeting the 12 years and above population* <https://www.afro.who.int/news/ethiopia-launches-covid-19-vaccination-campaign-targeting-12-years-and-above-population> (2023).
2. Gudina, E. K. *et al.* Seroepidemiology and model-based prediction of SARS-CoV-2 in Ethiopia: longitudinal cohort study among front-line hospital workers and communities. *en. Lancet Glob Health* **9**, e1517–e1527 (Nov. 2021).
3. Merkt, S. *et al.* *Supplementary files to Long-term monitoring of SARS-CoV-2 seroprevalence and variants in Ethiopia provides prediction for immunity and cross-immunity* Aug. 2023. <https://doi.org/10.5281/zenodo.8270192>.
4. Leinonen, R., Sugawara, H., Shumway, M. & International Nucleotide Sequence Database Collaboration. The sequence read archive. *en. Nucleic Acids Res.* **39**, D19–21 (Jan. 2011).
5. Sisay, A. *et al.* Molecular Epidemiology and Diversity of SARS-CoV-2 in Ethiopia, 2020–2022. *en. Genes* **14** (Mar. 2023).
6. Pedregosa, F. *et al.* Scikit-learn: Machine Learning in Python. *Journal of Machine Learning Research* **12**, 2825–2830 (2011).
7. Stein, C. *et al.* Past SARS-CoV-2 infection protection against re-infection: a systematic review and meta-analysis. *Lancet* **401**, 833–842 (Mar. 2023).

REVIEWER COMMENTS

Reviewer #1 (Remarks to the Author):

Merkt et. al. have continued to improve the manuscript and put in the effort to address the review comments. Despite the improvement of the manuscript, I still have a few comments that I would appreciate feedback on. In particular, I find that they need to critically review their interpretation of the results.

- Line 52 - suggests that increased early vaccination could have substantially reduced infections during the delta and omicron waves – The study area is only Addis Ababa and Jimma of Ethiopia, but the authors make conclusions regarding the entire country. Can they provide their basis for these assertions? How comparable are the other parts of the country to the study area?

- Line 112 – Could the authors clarify what they mean by, “combined time-resolved Anti-N and Anti-S antibody dataset”? How was the time resolution of the dataset performed?

- Line 144 - is the positive test for both anti-S and anti-N or either of the two? Is there a chance that there could just be cross-reactivity? For example, certain populations in the continent have been found to possess antibodies even though evidence of prior infection could not be determined.

- Line 154 – 157 – The authors state, “most individuals vaccinated also showed reactivity for Anti-N (in Round 5: 94.8% of the healthcare workers and 96.4% of the community members), suggesting they had been exposed to the infection before or shortly after vaccination.” Isn’t the goal of vaccination to produce immunity (antibodies)? Could they clarify why the reactivity observed is not attributed to the vaccination but rather to previous or new infection?

- Line 191 -194 – the authors claim, that the high levels of both anti-N and anti-S is associated with two or more infections. I find this a bit hard to understand. Is it that only one of the two can be induced in a given infection? Could it be possible then that for the remaining 25%, if they had a second infection then they still produced the same antibodies? Anti-S or anti-N as? And for those who had high levels of both, would this then be attributed to co-infections or early dual infections? Appreciably, there is a larger percentage with both in the later stage, but can this also be attributed to the combination of prior infection and vaccination? Furthermore, considering Figure 1 F and G, a similar trend is observed between both sets of antibodies suggesting that they were accumulated at the same rate and therefore both may be induced on the same infection.

- Line 239 – 240, “Figure 2e demonstrates that our dataset encompasses a range of distances up to 6, indicating diverse genetic distances between the variants.” The relevance of this statement is unclear. Instead, the authors could emphasize the continued introduction of new variants that expanded the genetic diversity in the country over time since we know that these VOCs did not emerge in Ethiopia and these were not the products of the continued evolution of the wild-type virus.

- Line 399 -341, “The analysis of the model predicted that the most common pathway of infections and vaccinations was: 1st infection with wildtype, 2nd infection with delta, vaccination, and 3rd infection with omicron BA.4/5”, although true, the statement appears to be redundant as this is also the order of occurrence of the pathways of infection. It is unclear the need to mention. It may be more useful for the authors to highlight and discuss the deviation from this pattern as the main point for this section.

- Line 589-590, The authors claim “we concluded that it would have been possible to substantially mitigate the delta and omicron waves with more administered vaccines.” However, these variants, especially Omicron is associated with high immune escape and was able to easily penetrate even in populations with high vaccination levels such as Europe and UK, due to the additional mutations acquired that allowed them to escape both innate and vaccine immunity. There is some evidence of additional protection from vaccination (<https://ukhsa.blog.gov.uk/2022/02/10/how-well-do-vaccines-protect-against-omicron-what-the-data-shows/>), but even at 3 does, only an efficacy of

71% could be achieved, which quickly waned to 46% in a few months. Could the authors explain how this would be possible?

- Line 646, given the sample space of the study, I suggest the authors perhaps rephrase the sentence, "revealed that over 96% were exposed at least once to the virus by the last round of our survey." Limiting it to the study population and not extrapolating to the national level.

- In the spirit of reproducibility, I applaud the authors for sharing the model data and yaml files on Zenodo. Would they also include a GitHub repository with all the scripts and steps to reproduce all the analyses and figures generated?

Reviewer #1 (Remarks on code availability):

The code was not provided by the authors.

Reviewer #2 (Remarks to the Author):

The authors have provided a thorough response to my comments.

One small point: in the the new figure where antibody responses are stratified by site (Figure SN4new), I don't see any vaccinated individuals. Is this omission intentional?

Reviewer #3 (Remarks to the Author):

The authors have addressed all my previous concerns and I have no further comments

Author's response to reviewers' comments

We thank the reviewers for their constructive comments. Based on their remarks, we have revised the manuscript. We addressed all their comments and tried to resolve the raised issues. Our point-by-point response is provided below.

The teal boxes contain the reviewers' comments, while the yellow boxes contain the changes made in the manuscript. To provide some context, the changes (blue) are surrounded by unchanged text (black).

Response to Reviewer #1

Merkt et. al. have continued to improve the manuscript and put in the effort to address the review comments. Despite the improvement of the manuscript, I still have a few comments that I would appreciate feedback on. In particular, I find that they need to critically review their interpretation of the results.

We thank the reviewer for the feedback to our revision. We addressed the comments below by highlighting the two region scope of our study and findings early on, adding more details on the vaccinations' mechanism, discussing the immunity escape properties of later variants more thoroughly and including other clarifications throughout the manuscript.

Line 52 - suggests that increased early vaccination could have substantially reduced infections during the delta and omicron waves – The study area is only Addis Ababa and Jimma of Ethiopia, but the authors make conclusions regarding the entire country. Can they provide their basis for these assertions? How comparable are the other parts of the country to the study area?

We apologize for the briefness of phrasing. In the Discussion section we already consider representability of our results. Now we made changes to the abstract clarifying already here, that antibody and PCR-test data, hence most of our findings, only come from two regions in Ethiopia, while still remaining inside the 150 words limit.

Changes in the manuscript:

Initial submission:

Under-reporting of COVID-19 cases and the lack of information about circulating SARS-CoV-2 variants remain major challenges for many African countries. Here, we present a comprehensive analysis of the SARS-CoV-2 infection dynamics in Ethiopia, focusing on reinfection dynamics, (variant-specific) immunity, and the impact of vaccination rates. We conducted an antibody serology study, sequenced PCR-test positive samples, used available test positivity rates, and constructed two mathematical models integrating this data. A multivariant model explores the variant dynamics identifying wildtype, alpha, delta, and omicron BA.4/5 as the most important variants in Ethiopia. Cross-immunity between variants is investigated, revealing immunities ranging from 24% to 69% risk reduction. An antibody-level focused model predicts slow antibody decay leading to sustained high antibody levels until present. Retrospectively, it suggests that increased early vaccination could have substantially reduced infections during the delta and omicron waves. However, further vaccination is less likely to have a significant impact.

Revised submission:

Under-reporting of COVID-19 and **limited** information about circulating SARS-CoV-2 variants remain major challenges for many African countries. **We analyzed** SARS-CoV-2 infection dynamics **in Addis Ababa and Jimma**, Ethiopia, focusing on reinfection, immunity, and the vaccination **effects**. We conducted an antibody serology study, sequenced PCR-test positive samples, used available test positivity rates, and constructed two mathematical models integrating this data. A multivariant model explores the variant dynamics identifying wildtype, alpha, delta, and omicron BA.4/5 as the important variants in **the study population**. Cross-immunity between variants is investigated, revealing immunities ranging from 24% to 69% risk reduction. An antibody-level focused model predicts slow antibody decay leading to sustained high antibody levels until present. Retrospectively, it suggests that increased early vaccination could have substantially reduced infections during the delta and omicron waves **in the considered group of individuals**. However, further vaccination is less likely to have a significant impact.

Line 112 – Could the authors clarify what they mean by, “combined time-resolved Anti-N and Anti-S antibody dataset”? How was the time resolution of the dataset performed?

“Time-resolved” here refers to the fact that each of the five rounds of sampling per site was conducted in a different time-span as indicated in Figure 1. We adjusted the manuscript to clarify this.

Changes in the manuscript:

Initial submission:

Furthermore, leveraging the combined time-resolved Anti-N and Anti-S antibody dataset, we conducted a detailed temporal analysis, comparing the antibody levels observed during the initial three rounds with those from the subsequent two rounds.

Revised submission:

Furthermore, **we leveraged the information from multiple rounds of sampling, which provided Anti-N and Anti-S antibody levels of individuals**. **The resulting dataset was used for** a detailed temporal analysis, comparing the antibody levels observed during the initial three rounds with those from the subsequent two rounds.

Line 144 - is the positive test for both anti-S and anti-N or either of the two? Is there a chance that there could just be cross-reactivity? For example, certain populations in the continent have been found to possess antibodies even though evidence of prior infection could not be determined.

Although it is possible that antibodies obtained from seasonal corona viruses get activated during a SARS-CoV-2 infection, the antibody tests we use are highly specific to SARS-CoV-2 antibodies^{1,2}. We included the following clarification into the manuscript:

Changes in the manuscript:

Initial submission:

Our analysis revealed that in April 2022 the majority of individuals (in Round 5: 95.9% of the healthcare workers and 94.8% of the community members), tested positive for Anti-S and Anti-N antibodies (Figure 1a–e), suggesting an infection event.

Revised submission:

Our **SARS-CoV-2 specific antibody tests** revealed that in April 2022 the majority of individuals (in Round 5: 95.9% of the healthcare workers and 94.8% of the community members), **reacted** positive for **both** Anti-S and Anti-N antibodies (Figure 1a–e), suggesting an infection event. **Based on a previous study this result is unlikely to be explained by cross-reactivity².**

Line 154 – 157 – The authors state, “most individuals vaccinated also showed reactivity for Anti-N (in Round 5: 94.8% of the healthcare workers and 96.4% of the community members), suggesting they had been exposed to the infection before or shortly after vaccination.” Isn’t the goal of vaccination to produce immunity (antibodies)? Could they clarify why the reactivity observed is not attributed to the vaccination but rather to previous or new infection?

The two vaccines making up the overwhelming majority of doses administered in Ethiopia during our study period are Covshield (AstraZeneca type vaccine manufactured by Serum Institute of India) and Johnson and Johnson. The former “consists of a replication-deficient chimpanzee adenoviral vector ChAdOx1, containing the SARS-CoV-2 structural surface glycoprotein antigen (spike protein; nCoV-19) gene”³. Also the latter “is made up of another virus (an adenovirus) that has been modified to contain the gene for making the SARS-CoV-2 spike protein”⁴. Hence the immune system responds to the vaccines by producing antibodies specific only to the spike protein (Anti-S). In contrast the response to the actual SARS-CoV-2 consists of antibodies against various targets – amongst others against the nucleocapsid (Anti-N) and spike protein for which we tested. This allows us to distinguish between previous vaccination and infection by checking if the study participant is positive only for Anti-S (vaccine) or for both (infection). We made the following clarifications in the manuscript:

Changes in the manuscript:

Initial submission:

Interestingly, most individuals vaccinated also showed reactivity for Anti-N (in Round 5: 94.8% of the healthcare workers and 96.4% of the community members), suggesting they had been exposed to the infection prior to or shortly after vaccination.

Revised submission:

Interestingly, **although the vaccines used in Ethiopia only induce Anti-S**, most individuals vaccinated also showed reactivity for Anti-N (in Round 5: 94.8% of the healthcare workers and 96.4% of the community members), suggesting they had been exposed to the infection prior to or shortly after vaccination.

Line 191 -194 – the authors claim, that the high levels of both anti-N and anti-S is associated with two or more infections. I find this a bit hard to understand. Is it that only one of the two can be induced in a given infection? Could it be possible then that for the remaining 25%, if they had a second infection then they still produced the same antibodies? Anti-S or anti-N as? And for those who had high levels of both, would this then be attributed to co-infections or early dual infections? Appreciably, there is a larger percentage with both in the later stage, but can this also be attributed to the combination of prior infection and vaccination? Furthermore, considering Figure 1 F and G, a similar trend is observed between both sets of antibodies suggesting that they were accumulated at the same rate and therefore both may be induced on the same infection.

The occurrence of high levels of both Anti-N and Anti-S antibodies indeed suggests multiple exposures to the virus. It is plausible that in some cases, individuals produce predominantly one type of antibody in response to a particular variant or due to the timing of the sample post-infection. However, our data also suggest that most individuals eventually develop both Anti-N and Anti-S antibodies upon repeated exposures, aligning with the notion of a boosted immune response. As for the participants with only one type of high-level antibody, it may reflect recent vaccination or an infection, where the immune response has not fully matured to reflect dual high levels. The observed trend of increasing percentages of individuals with both high Anti-N and Anti-S in later stages may indeed reflect cumulative exposures to infections and/or vaccinations. To clarify these points, we made the following modifications to the manuscript:

Changes in the manuscript:

Initial submission:

Remarkably, in the latest round of sample collection in April 2022, a substantial proportion (75-80%) of the sampled individuals exhibited high antibody levels for Anti-N as well as Anti-S. This suggests that a significant fraction of the population had already experienced at least two exposures for each antigen by that time.

Revised submission:

Remarkably, in the latest round of sample collection in April 2022, a substantial proportion (75-80%) of the sampled individuals exhibited high antibody levels for Anti-N as well as Anti-S. Since Anti-N is only induced after an infection due to the spike-protein nature of the vaccines used in Ethiopia, this suggests that a significant fraction of the population had already experienced at least two exposures for each antigen by that time.

Line 239 – 240, “Figure 2e demonstrates that our dataset encompasses a range of distances up to 6, indicating diverse genetic distances between the variants.” The relevance of this statement is unclear. Instead, the authors could emphasize the continued introduction of new variants that expanded the genetic diversity in the country over time since we know that these VOCs did not emerge in Ethiopia and these were not the products of the continued evolution of the wild-type virus.

We appreciate the reviewer’s suggestion and agree that emphasizing the introduction of new variants and the resulting genetic diversity is additionally informative. The statement regarding the range of genetic distances was intended to highlight the diversity of the virus in Ethiopia and in particular put it into terms of “distance in VOC” since this is later used for modelling. It is now accompanied by the context of variant introduction and evolution.

Changes in the manuscript:

Initial submission:

Grouping our variants by MOIC allows us to maintain statistical power of the lineage groups for subsequent analysis of potential cross-immunity while still retaining their relevant spike protein differences. The grid of distances of MOIC between observed lineage groups in Figure 2e demonstrates that our dataset encompasses a range of distances up to 6, indicating diverse genetic distances between the variants.

Revised submission:

Grouping our variants by MOIC allows us to maintain statistical power of the lineage groups for subsequent analysis of potential cross-immunity while still retaining their relevant spike protein differences. The grid of distances of MOIC between observed lineage groups in Figure 2e demonstrates that our dataset encompasses a range of distances up to 6, indicating diverse genetic distances between the variants. **Moreover we see that our data set consists of variants which emerged earlier in other parts of the world, hence implies a continuous introduction of new variants to Ethiopia rather than a mutation of the wild-type inside of Ethiopia.**

Line 399 -341, “The analysis of the model predicted that the most common pathway of infections and vaccinations was: 1st infection with wildtype, 2nd infection with delta, vaccination, and 3rd infection with omicron BA.4/5”, although true, the statement appears to be redundant as this is also the order of occurrence of the pathways of infection. It is unclear the need to mention. It may be more useful for the authors to highlight and discuss the deviation from this pattern as the main point for this section.

We checked the text and could not find any other appearance of this information. It was intended to summarize the predominant infection pathway and explaining Figure 4. However, we agree that highlighting deviations from this pathway and discussing their implications could provide more insightful and novel information adjusted the manuscript accordingly.

Changes in the manuscript:

Initial submission:

The analysis of the model predicted that the most common pathway of infections and vaccinations was: 1st infection with wildtype, 2nd infection with delta, vaccination, and 3rd infection with omicron BA.4/5 (Figure 4a,b).

Revised submission:

The analysis of the model predicted that the most common pathway of infections and vaccinations was: 1st infection with wildtype, 2nd infection with delta, vaccination, and 3rd infection with omicron BA.4/5 (Figure 4a,b). **In particular wildtype*, alpha, beta, eta, omicron BA.1 are not part of it, of which omicron BA.1 appears in the second most common pathway (delta, omicron BA.1, omicron BA.4/5) and alpha appears in the third most common pathway (alpha, delta, vaccination, omicron BA.4/5).**

Line 589-590, The authors claim “we concluded that it would have been possible to substantially mitigate the delta and omicron waves with more administered vaccines.” However, these variants, especially Omicron is associated with high immune escape and was able to easily penetrate even in populations with high vaccination levels such as Europe and UK, due to the additional mutations acquired that allowed them to escape both innate and vaccine immunity. There is some evidence of additional protection from vaccination (<https://ukhsa.blog.gov.uk/2022/02/10/how-well-do-vaccines-protect-against-omicron-what-the-data-shows/>), but even at 3 does, only an efficacy of 71% could be achieved, which quickly waned to 46% in a few months. Could the authors explain how this would be possible?

This is a crucial point. The effectiveness of vaccination in the face of variants with high immune escape potential, like Omicron, is indeed a complex issue. The cited efficacy rates and the observed rapid waning of protection emphasize the need for a nuanced discussion of vaccination’s role regarding the different variant. In our manuscript, we argue for the potential of vaccination to mitigate the impact of these waves, not to prevent them entirely. Moreover, the model allows for different infectiousness parameters for the different variants and indeed for later variants we estimated higher values, implying a higher immune escape potential. We now included information about this also to the result section of the main manuscript and discuss in more detail the differences of the effect on delta and omicron and in particular the uncertainties of the omicron wave.

Changes in the manuscript:

Initial submission:

Indeed, credible intervals for parameter estimates (Supplementary Information Table SN6), state variables (Figure 5b) and predictions (Figure 5c) were mostly tight, indicating a low uncertainty of model predictions.

...

Furthermore, by simulating higher vaccination rates retrospectively, we concluded that it would have been possible to substantially mitigate the delta and omicron waves with more administered vaccines. This is strongly supported by our healthcare worker antibody data, where in August 2021 most of the high antibody levels were caused by vaccination in comparison to community members with almost no vaccination, but similarly high level percentages (Figure 1d, Supplementary Figure 3d). However, from then on most of the population was exposed multiple times and thus benefits of the titres are less pronounced now.

Revised submission:

Indeed, credible intervals for parameter estimates (Supplementary Information Table SN6), state variables (Figure 5b) and predictions (Figure 5c) were mostly tight, indicating a low uncertainty of model predictions. In alignment with immune escape properties of later variants, we estimated higher valued infectiousness parameters for them, e.g. omicron BA.4/5 having 3.3 times the delta and 10.6 times the wildtype infectiousness. Relative infectiousness for all variants can be deduced from Supplementary Note Tables SN3 and SN6.

...

Furthermore, by simulating higher vaccination rates retrospectively, we concluded that it would have been possible to substantially mitigate the delta and omicron waves with more administered vaccines. For the delta wave this is strongly supported by our healthcare worker antibody data, where in August 2021 most of the high antibody levels were caused by vaccination in comparison to community members with almost no vaccination, but similarly high level percentages (Figure 1d, Supplementary Figure 3d). On the other hand, for omicron we have high uncertainties in our predictions (Figure 5c). Taking into account the high immune escape property of omicron we would probably still have seen a substantial wave, nevertheless with a notably smaller peak. Moreover, from then on most of the population was exposed multiple times and thus benefits of the titres are less pronounced now.

Line 646, given the sample space of the study, I suggest the authors perhaps rephrase the sentence, “revealed that over 96% were exposed at least once to the virus by the last round of our survey.” Limiting it to the study population and not extrapolating to the national level.

We appreciate the reviewer’s attention to the representativeness of our findings. It is important to ensure that our conclusions are not overgeneralized beyond the scope of our study population and we changed the manuscript accordingly.

Changes in the manuscript:

Initial submission:

However, our five rounds of seroepidemiological survey in Addis Ababa and Jimma between August 2020 and April 2022, revealed that over 96% were exposed at least once to the virus by the last round of our survey.

Revised submission:

However, our five rounds of seroepidemiological survey in Addis Ababa and Jimma between August 2020 and April 2022, revealed that **in our study group** over 96% were exposed at least once to the virus by the last round of our survey.

In the spirit of reproducibility, I applaud the authors for sharing the model data and yaml files on Zenodo. Would they also include a GitHub repository with all the scripts and steps to reproduce all the analyses and figures generated?

We opted for Zenodo over GitHub in order to comply with Nature Communication's policy of citing if possible only sources with a DOI and to guarantee long-term sustainability independently of private companies. In addition to the already available model creation code we now also uploaded the figure creation code to Zenodo (<https://doi.org/10.5281/zenodo.8313899>).

The code was not provided by the authors.

The code for model creation had been provided as referenced in Zenodo (<https://doi.org/10.5281/ZENODO.8270192>). We now also included the code for figure creation (<https://doi.org/10.5281/zenodo.8313899>).

Response to Reviewer #2

The authors have provided a thorough response to my comments.

We thank the reviewer for this positive evaluation of our revised manuscript.

One small point: in the the new figure where antibody reponses are stratified by site (Figure SN4new), I don't see any vaccinated individuals. Is this ommission intentional?

We changed the figure to account for vaccinated individuals.

Response to Reviewer #3

The authors have addressed all my previous concerns and I have no further comments

We thank the reviewer for this positive evaluation of our revised manuscript.

Response Letter References

1. Eser, T. M. *et al.* Nucleocapsid-specific T cell responses associate with control of SARS-CoV-2 in the upper airways before seroconversion. *Nature Communications* **14**, 2952. ISSN: 2041-1723. <https://doi.org/10.1038/s41467-023-38020-8> (2023).

2. Olbrich, L. *et al.* Head-to-head evaluation of seven different seroassays including direct viral neutralisation in a representative cohort for SARS-CoV-2. *Journal of General Virology* **102**. <https://www.microbiologyresearch.org/content/journal/jgv/10.1099/jgv.0.001653> (2021).
3. Voysey, M. *et al.* Safety and efficacy of the ChAdOx1 nCoV-19 vaccine (AZD1222) against SARS-CoV-2: an interim analysis of four randomised controlled trials in Brazil, South Africa, and the UK. *Lancet* **397**, 99–111 (2021).
4. EMA. *EMA receives application for conditional marketing authorisation of COVID-19 Vaccine Janssen* <https://www.ema.europa.eu/en/news/ema-receives-application-conditional-marketing-authorisation-covid-19-vaccine-janssen> (2024).